# Frequency-selective actuation of liquid crystalline elastomer actuators with radio-frequency

Yiwen Song [1,6] ✉, Zefang Li [2,3,6], Mason Zadan [2,4], Jingxian Wang [1,5], Swarun Kumar [1] ✉ & Carmel Majidi [1,2] ✉

Soft and miniaturized robots possess the capability to operate inside narrow, confined environments. However, powering soft robots inside these environments with on-board batteries or wired connections to external power supplies can significantly restrain their mobility. Similarly, wireless actuation approaches are constrained by near-field actuation, line-of-sight operation, or indiscriminate actuation of many actuators. To provide higher mobility for wireless soft robot to operate inside non-line-of-sight scenarios, we present a radio-frequency system that introduces frequency-selective actuation of liquid crystal elastomer actuators. We create liquid crystalline elastomer actuators with a low actuation temperature and embed them with conductive traces that resonate and heat by selected frequencies of radio-frequency excitation in the 2.40 GHz range. We further develop a wireless actuation platform that infers the wireless channel and beamforms towards the actuator to achieve efficient beamforming. Demonstrations show our system is capable of selectively actuating different actuators while the robot is in motion and obstructed by occlusions.

Advances in soft robotics have simplified robot-environment interactions and enabled smaller and lighter field robots that can interact and deform through their environments much more aptly. In order to match the mobility and maneuverability of natural organisms, soft robots need to be lightweight and fully untethered. Early efforts in mobile, untethered soft robotics were successful in demonstrating biologically relevant modes of locomotion using a variety of actuation methods, including pneumatic actuators[1–3], dielectric elastomers[4–6], and shape memory alloy[7–9]. While promising, these robot designs were limited in speed and mobility due to the size and weight of on-board compressed air sources, high voltage controllers, or batteries, respectively.

To combat the limitations caused by tethered robots, efforts have focused on untethered soft robots. Soft robots that carry on on-board batteries[10–12] are restrained by the size and weight of batteries. To achieve battery-free actuation, researchers have designed actuators that can be deformed under magnetic fields[13–15] and through inductive heating with coils[16]. These near-field actuation methods are still constrained by the short operating distance, where the driven magnet and coils need to be placed a few millimeters to centimeters away from the actuator. Light sources such as laser[17–19] and near-infrared[20–24] actuation can achieve actuation in distance, but they can only operate in line-of-sight. This property not only restrains the operating environment of the robot, but also restricts the robotic design, as the actuators must be externally exposed.

This paper builds on our own preliminary work exploring microwave heating as a driving mechanism for soft robots[25,26]. Radio-frequency is an ideal source for remotely stimulating actuation since it

[1]Department of Electrical and Computer Engineering, Carnegie Mellon University, Pittsburgh, Pennsylvania, US. [2]Department of Mechanical Engineering, Carnegie Mellon University, Pittsburgh, Pennsylvania, US. [3]Present address: Department of Civil and Systems Engineering, Johns Hopkins University, Baltimore, Maryland, US. [4]Present address: Koch Institute for Integrative Cancer Research, Massachusetts Institute of Technology, Cambridge, Massachusetts, US. [5]Present address: Computer Science Department, National University of Singapore, Singapore, Singapore. [6]These authors contributed equally: Yiwen Song, Zefang Li. ✉e-mail: yiwens2@andrew.cmu.edu; swarun@cmu.edu; cmajidi@andrew.cmu.edu

can penetrate through occlusions. Prior work has investigated the feasibility of using microwave power to directly heat elastomers through dielectric heating[27,28] or heat ferromagnetic alloys through inductive heating[29–31]. These systems require more than 100 watts of power and tens of seconds to heat elastomers up to 60 °C within a 10 cm range as they blindly blast power into the space. Our most recent work has shown that through wireless beamforming, soft actuators such as liquid crystalline elastomer (LCE) with a conductive layer can be heated more efficiently at a higher speed, even in the presence of occlusions[25,26]. However, due to the large wavelength of radio frequencies, these previous systems could not be designed to support independent control of multiple actuators or in compact antenna size

configurations, making it impossible for structured coordinated motions between actuators.

In this paper, we introduce RFact. RFact is a soft robotics platform that achieves efficient and selective actuation for battery-free thermally driven soft robotic locomotion. It can operate within a range of 30 cm, even with the presence of non-metallic obstructions. RFact achieves this through a combined design of LCE chemistry, soft actuator integration design, and a radio-frequency beamforming system, shown in Fig. 1. First, RFact introduces a chemistry for an LCE actuator that responds to relatively low temperature stimulation to fit the context of wireless actuation with less power. Second, RFact exploits the frequency-selective actuation capability of the LCE-based

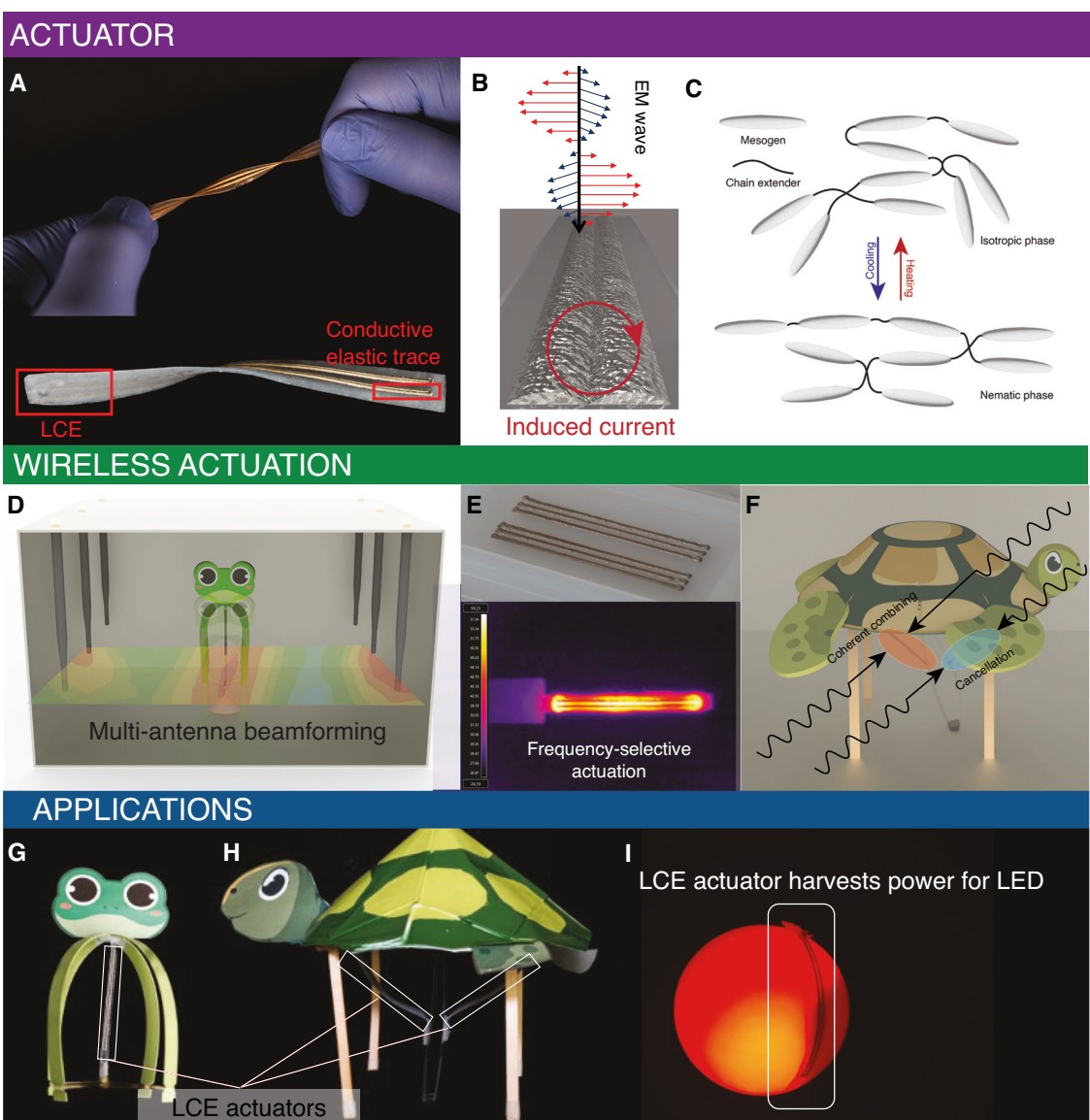

**Fig. 1 | Overview. A** LCE-based actuator coated with soft conductive ink for absorbing electromagnetic power and enabling frequency-selective heating and actuation. **B** EM waves interact with the conductive elastic patterns, causing induced current to form on the surface and enabling inductive heating. **C** On the micro levels, the mesogens, which are the high-aspect rigid monomers of the LCE structures, experience phase transition and become disoriented while being heated, causing the LCE actuator to contract macroscopically. **D** The wireless actuation platform consists of six antennas that perform multi-antenna distributed beamforming toward target locations. **E** Parallel resonating LM-based conductive patterns that are 3D-printed onto the actuator can achieve efficient frequency-

selective heating based on varying trace dimensions. **F** Beamforming achieved by the actuation system can focus wireless power at the desired regions and null the wireless power at undesired regions. **G** A jumper robot is actuated wirelessly by the actuator. We beamform wireless power to the actuator and heat the actuator to over 120 °C before the stored mechanical energy is released and the robot jumps. **H** A crawling robot enabled by two separate actuators. The wireless power is beamformed and frequency controlled to actuate each actuator separately to enable forward and backward locomotion of the soft robot. **I** The actuators are shown here to double as passive energy harvesters during actuation with the potential to power onboard systems.

actuators. This is achieved by integrating LCE with 3D-printed conductive traces containing eutectic gallium indium (EGaIn) liquid metal (LM) that are tuned to resonate at different frequencies based on control of the trace dimensions and surface roughness. The soft actuator design allows for RF heating as well as harvesting power, treating the conductive LM-based actuators as antennas for low-power electronics. Third, RFact proposes a frequency-aware channel interpolation algorithm for beamforming towards a desired actuator location. The frequency-aware beamforming allows actuation at different places and different frequencies through measuring the channel at a few locations in advance.

RFact serves as the first proof-of-concept system towards a wireless actuation platform that achieves a comparable range of mobility as soft robots with on-board batteries[32,33] - e.g., crawling, hopping, picking up objects, and power harvesting to power electronics. Although the focus of RFact is not in producing robotic systems to accomplish complex tasks, the demonstration of RFact opens opportunities for using wireless power to actuate soft robots in future work.

## Results
RFact proposes a wireless actuation system that achieves higher efficiency and high selectivity in energy transfer and actuator selection. This efficiency is achieved by designing thermal transducers made from shape memory materials (SMP) that actuate at lower temperatures, along with a wireless multi-antenna beamforming system. The selectivity is achieved by controlling the design parameters of the frequency-selective LCE and LM-based actuators and spatially frequency-aware beamforming.

Thermally responsive LCEs typically have phase transition temperatures above 70 °C[34], which requires a large amount of power to achieve full actuation. To actuate LCE actuators towards such a high temperature, high RF input power is needed to heat over a small region with more miniature actuators. To accomplish this, we aim to activate the LCE actuators with a similar amount of power used in our previous work, but with LCE actuators that are of a smaller size over a smaller area. To achieve this, RFact aims at designing an LCE actuator that can produce a larger amount of force output and shape change at a lower and tunable temperature. The tunable temperature allows users to choose between high actuation selectivity (cooperate with frequency-selective actuation) and low-power actuation. In particular, we present a method to selectively mix monomers to weaken the interaction between chain bonds by looking into the chemical mechanism of LCE actuation. Specifically, we use mixed monomers to weaken the $\pi - \pi$ interaction in the LCE at the nematic phase. Second, we study the resonance of 3D-printed conductive traces. Our past works have shown that using LM coating or conductive-trace coating on LCE can significantly improve actuation speed due to induction heating. This work conducts an extensive study on how the design and fabrication of conductive traces would affect the resonant frequency and heating speed. We then package the LCE with the 3D-printed conductive traces to fabricate actuators that are frequency-selective. Frequency-selective actuators in close spatial regions can be selectively actuated.

Finally, we study the frequency-aware beamforming. Beamforming is achieved through a network of multiple antennas that coherently combines power towards desired locations. The frequency-selective actuation creates new challenges for beamforming since the wireless channel changes with frequencies. We study the standing waves inside the environment and use frequency-aware interpolation to create beamforming weights corresponding to the current operating frequency and the location to be beamformed. The method allows few-shot probes of the channel before beamforming.

### Low $T_{ni}$ liquid crystal elastomer
Typical LCEs based on liquid crystal monomers such as RM257 (1,4-bis-[4-(3-acryloyloxypropyloxy) benzoyloxy]-2-methylbenzene)

have nematic-isotropic phase transition temperatures ($T_{ni}$) ranging from around 66 °C to above 100 °C[35]. To reduce the phase transition temperature of LCEs, we adopt a formulation with weaker bonds that maintain structural integrity and enable shape transitions at a temperature of approximately 50 °C. To reduce the $T_{ni}$ of LCE, two approaches have been adapted: altering the LCE network and changing the aliphatic chain lengths[36,37] or replacing a portion of liquid crystal diacrylates with PEG[37] or with another liquid crystal diacrylate with significantly lower $T_{ni}$[38]. In prior efforts that have led to significant reductions in $T_{ni}$, either flexible PEG chains are added to the LCE, which tunes the chemistry toward elastomeric properties (as opposed to liquid crystal elastomer)[37], or some monomers with different conjugate systems, which can weaken the $\pi - \pi$ interaction in LCE at the nematic phase, are added (TATATO[36], and C6BAPE[38]). Although previous papers have shown a reduction in phase transition temperature, our work is the first to show an LCE material that achieves tunable, lower phase transition temperature with a low compromise in the mechanical performance, which indicates an LCE recipe of higher thermal to mechanical power efficiency.

In this work, we introduce an alternative method to fluently control the $T_{ni}$ from 24 °C (room temperature) to the standard 75 °C of liquid crystal elastomers with a tunable range of co-monomer mixture ratio. Inspired by Bauman et al.[39], we propose to use co-monomer bisphenol A ethoxylate diacrylate to be mixed with liquid crystal monomer RM257 to synthesize LCEs. Replacing the RM257 monomer serves to weaken the inter-layer molecular bond and decrease the activation energy for the actuation reaction. Specifically, it is known that at elevated temperatures, the $\pi$-$\pi$ interaction between the mesogen cores is weakened and the mesogens gain more mobility and ability to transition to the isotropic phase. Substituting part of RM257 with Bisphenol A ethoxylate diacrylate (Mn = 512 g/mol, thereafter Bisphenol-512) monomer can effectively weaken the $\pi$-$\pi$ interaction within liquid crystal mesogens cores and reduce the barrier energy that must be overcome for phase transition, resulting in a reduction of $T_{ni}$. Additional information on synthesis can be found in the Methods Section.

The decrease in $T_{ni}$ compared to prior formulations is highlighted with differential scanning calorimetry (DSC). The phase transition temperatures for different LCE compositions with different percentages of RM257 replaced by Bisphenol A ethoxylate diacrylate are shown in Fig. 2C. The tunable low-phase transition temperature enables the actuators to fit different operating temperatures.

Next, dynamic mechanical analysis (DMA) is conducted for the composition in which 5 mol% of RM257 is replaced by Bisphenol-512. The DMA results are shown in Fig. 2D. A concentration of 5% Bisphenol A ethoxylate diacrylate is selected for its effective stroke length and transition temperature, which allows for operation in near room temperature environments. An LCE formulation with this composition starts exhibiting contraction when heated from room temperature and completes actuation at ~50 °C with 29.6% contraction in line with the stroke length of unaltered high $T_{ni}$ LCE chemistry. Moreover, it is similar to the stroke length of natural human muscle and approaches the theoretical maximum of 33.3% for a given 50% pre-stretch.

The DMA result of our LCE is compared with a regular LCE without any Bisphenol-512 monomers. Figure 2E shows that to achieve 10% strain, LCE without Bisphenol-512 needs above 65 °C, which is 35 °C higher than LCE with 5% Bisphenol-512. Further, LCE with 5% Bisphenol-512 reaches full actuation at around 50 °C, which is 40 °C less than LCE without Bisphenol-512.

To understand the mechanical performance of the LCE actuator under high cyclical loads, uniaxial mechanical testing was conducted. The LCE actuator is tested on a universal load frame (Model 5969; Instron) with 3 independent samples and 5 repeated experiments for each curve. The LCE actuator is tested under the heating and cooling process from room temperature (25 °C) to 100 °C.

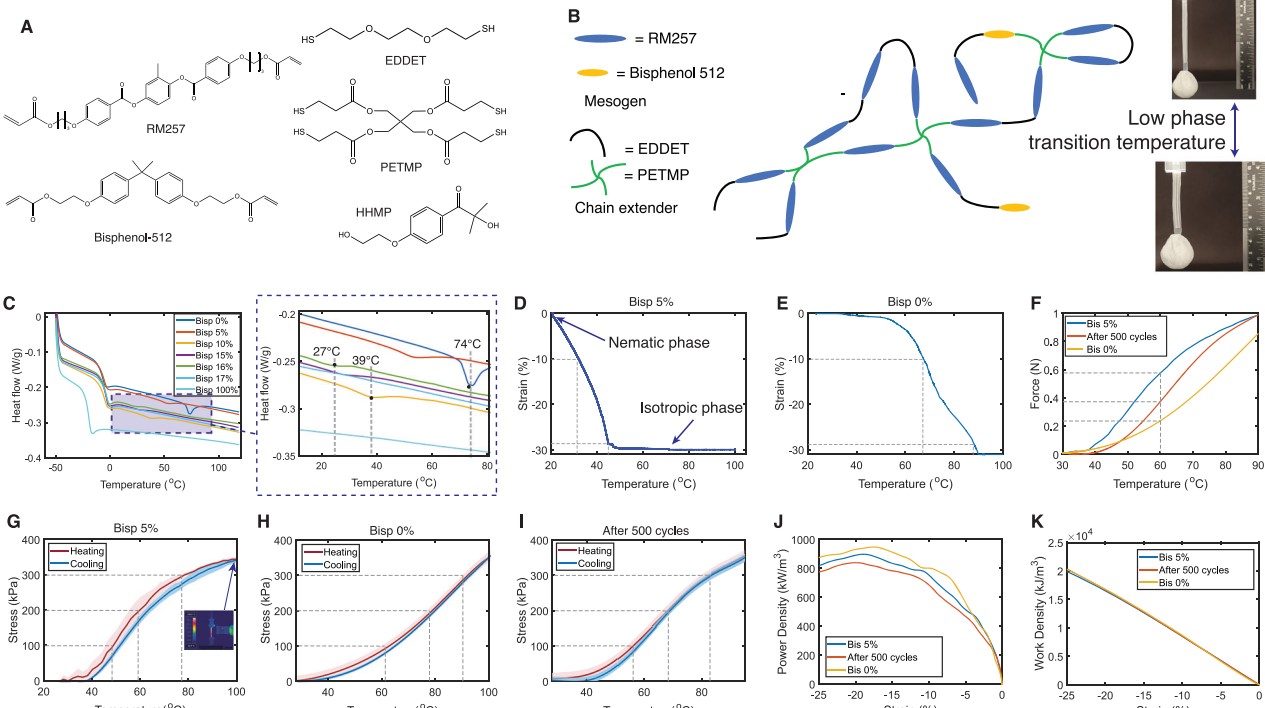

**Fig. 2 | Material characterization. A** Combination of Bisphenol-512 with RM257 mesogens allows for a low transition temperature LCE for improved wireless actuation performance. **B** Bisphenol-512 weakens the π-π stacking of RM257 chains, which causes a decrease in the transition temperature from the nematic phase to the isotropic phase. **C** Differential scanning calorimetry (DSC) testing of the LCE with different percentages of doped Bisphenol-512 into RM257 mesogen during the temperature rise process. Doping Bisphenol-512 significantly decreases the first phase transition temperature of the LCE. LCE with a larger than 17% Bisphenol-512 appears to be crystalline without nematic phase transitions. **D** Dynamic mechanical analysis (DMA) testing of the LCE with 5% of RM257 replaced by Bisphenol-512. **E** DMA of LCE without Bisphenol-512. **F** The force-temperature curve of actuators with 5% Bisphenol-512, actuators after 500 cycles, and an unmodified LCE actuator with 0% Bisphenol-512. **G–I** The stress-temperature curve of actuators with 5% Bisphenol-512, actuators (5% Bisp) after 500 cycles, and an unmodified LCE actuator with 0% Bisphenol-512. The red and blue shaded areas represent the area within one standard deviation. **J** The measured power density with a 0.5 N constant load, with regard to strain, for the three types of actuators. **K** The measured work density with a 0.5 N constant load, with regard to strain, for the three types of actuators.

The force output of three types of actuators (actuator with 5% Bisphenol-512, actuator (5% Bisp) after actuation of 500 cycles, and LCE with 0% Bisphenol-512) is tested, shown in Fig. 2F. The three actuators are cut to the same length with the same cross-sectional areas. LCE actuators with 5% Bisphenol-512 exhibit a shape memory response at a lower temperature, creating a higher force output at lower temperatures in the 30–90 °C interval. By dividing the force output by the cross-sectional areas, Fig. 2G–I shows the stress-temperature curve, where the shaded area represents error regions determined by one standard deviation. LCE actuators with 5% Bisphenol-512 provides 350 MPa stress from 25 °C to 100 °C. Compared to LCE actuators fabricated with pure RM257[25], the LCE actuator generates 2 times higher stress at 60 °C, and generates 1.5 times higher stress after 500 cycles.

The power and work density of the actuators are measured by applying a 0.5 N (equivalent to a 50 g hanging object) constant load to the actuator and heating it using a constant heat source, by a heat gun. Figure 2J, K shows the power and work density during the actuation period of the actuators, with regard to the strain of the actuator. This indicates that all actuators have a power density peak between -15% and -20% strain, achieving a maximum of 900 kW/m³ power density for our LCE. Although the power density of LCE with 5% Bisphenol-512 is slightly lower than LCE with 0% Bisphenol-512, and the power density after 500 cycles drops slightly, they still provide similar work densities, reaching a total of 20000 kJ/m³ at -25% strain. Since the actuators provide similar work density, the advantage of LCE with 5% Bisphenol-512 actuating at a lower temperature is that it provides opportunities for actuation with less heating time and power.

## RF Soft actuator design and characterization

RFact uses a 3D-printed elastic conductive pattern[40,41] embedded on the LCE layer that resonates with the RF energy and generates heat through induced heating, and thus heats the LCE to actuate. Such a conductive pattern is designed and tested to resonate in a tight frequency range. With multiple conductive patterns that resonate at different frequencies, selective actuation can be achieved. Images of a completed actuator are shown in Fig. 3A. The conductive trace is composed of silver flakes and eutectic gallium-indium (EGaIn) liquid metal blended within a soft polystyrene-block-polyisoprene-block-polystyrene (SIS) copolymer matrix. When suspended within a toluene solvent, this slurry enables precise direct-ink-writing (DIW) printing of the traces onto the LCE layer. Once the toluene has evaporated, LM-Ag-SIS stretchable traces remain with excellent conductivity under high loading, along with self-healing properties. See the Methods section for additional information on the formulation and materials background.

When stimulated with an RF beam, an electrical current is induced within the conductive film, and the film heats up due to Joule heating. The quarter-wavelength pattern is designed to resonate well within specific RF frequencies ranging from 2.35 GHz to 2.5 GHz, and the configuration is tested with our actuation testbed. The heating effect of the conductive patterns causes the LCE to heat up above its phase-transition temperature and undergo a shape memory actuator response. After the thermal stimulation is removed, the LCE cools down to room temperature and contracts back to its original shape.

Several variables affect the resonant frequency and heating efficiency of the LM-Ag-SIS conductive ink patterns. First, standing waves

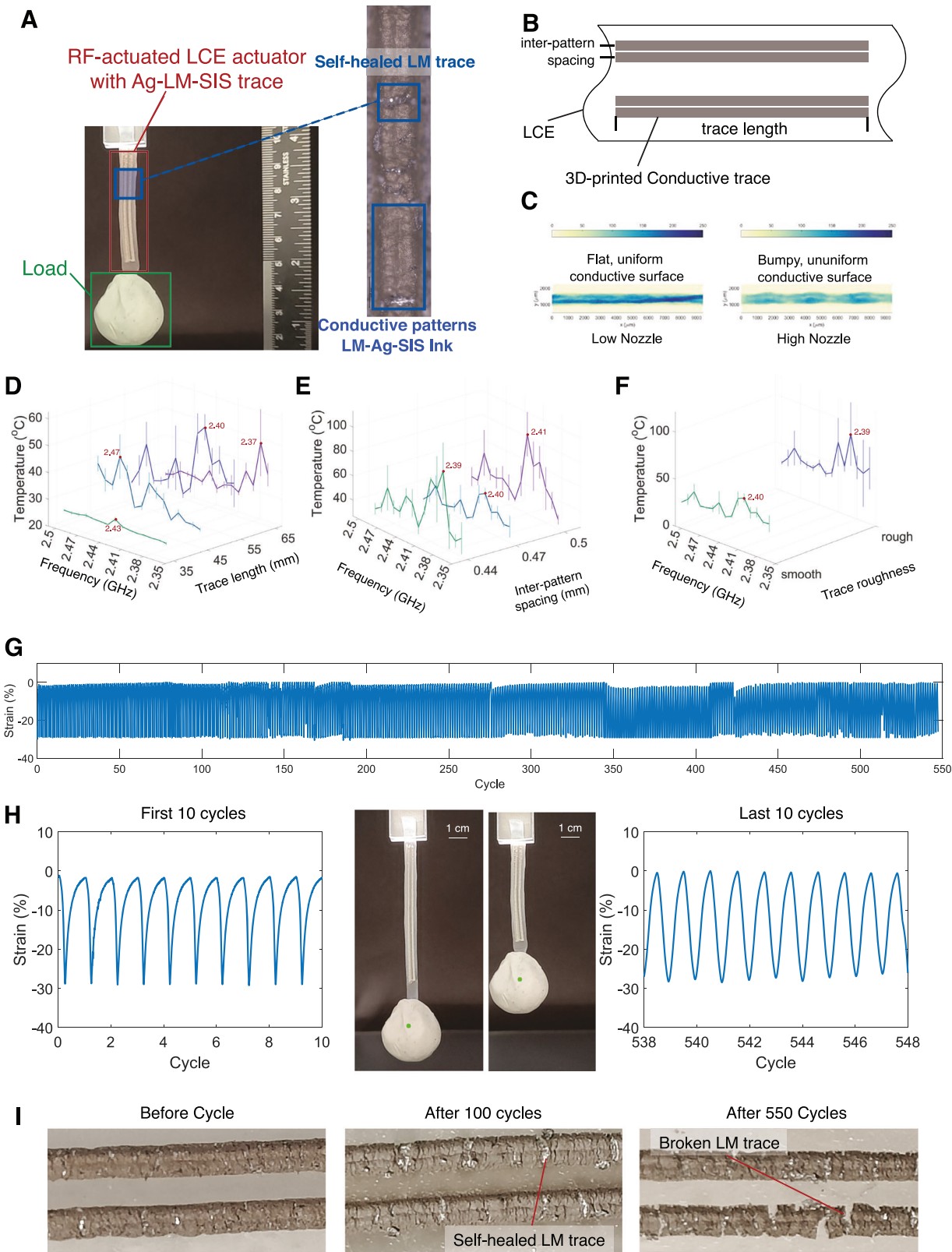

form on the conductive patterns while interacting with the RF waves, which are relevant to the length of the conductive patterns. The amplitude of standing waves formed on the conductive patterns determines the overall heating power. The distance between the two thin parallel wires can also be controlled at different values to enable varying resonant frequencies and heating efficiency. Finally, the different roughness of the conductive patterns, which can be controlled

by setting different nozzle heights of the 3D printer during printing, can also affect heating efficiency due to a change in the resistivity of the patterns. To address this, these parameters were explored and characterized to improve actuator performance.

We test the resonant frequency and heating efficiency of the LCE actuators under different variables of the conductive patterns: length, roughness, and inter-pattern spacing. The three variables significantly

**Fig. 3 | Characterization of RFact's frequency-selective actuator. A** Microscopy of the actuator after the cycle test shows a self-healed LM trace after heating and stretching. **B** An illustration of the 3D-printed conductive patterns on the actuator. **C** Confocal microscopy of the conductive trace with different printing configurations, leading to different heating performance. Each pattern is heated for 60 seconds and their temperature is recorded. Separate plots can be found in Supplementary Information 2. **D** The length of the conductive traces affects the resonant frequency and speed of heating. In particular, longer traces provide better heating performance, while the length also affects the resonant frequency of the actuator, where the maximum heating efficiency is located. **E** The roughness of the conductive patterns affects the heating efficiency. Two actuators with different printing configurations, where they present different heating efficiency. **F** The resonant frequency and heating efficiency of patterns with different inner-strip distances. **G** Cyclical actuation test of more than 500 cycles at 30% strain indicates almost no performance degradation in mechanical performance. **H** The maximum strain achieved is similar in the first and last 10 cycles. Image of experimental setup and optical tracking used to measure stroke length for cyclical testing. **I** Conductive traces before, after 100 cycles, and the broken trace after more than 500 cycles.

affect the result. To prepare the test samples, we print conductive LM-Ag-SIS ink on acrylic substrates with the same patterns used on the LCE actuators. Acrylic is selected as the substrate in order to mitigate shape change. Each trace is stimulated with a 2.5 W received signal strength. Since we only consider the temperature and heating efficiency until the actuation temperature is reached, the test well reflects the difference in variable control. For each variable, five repeated tests are conducted to ensure repeatability and data consistency.

We first evaluate the different wireless heating effects by different lengths of the conductive patterns. From an antenna design perspective, antennas of different lengths resonate at different frequencies. At the resonant frequency, the standing wave that is formed on the conductive layer reaches the maximum value, resulting in the fastest heating effect. The conductive patterns are set to be of length 35, 45, 55, and 65 mm, and the results are shown in Fig. 3D, where each actuator is heated for 30 s. Results indicate that the maximum temperature is achieved at 65 mm for 2.37 GHz RF stimulation, where the actuator achieves an average of 45 °C. Results also show deviation in the resonant frequency of different actuators, as the optimal resonant frequency is 2.40 GHz for 55 mm-long actuators, 2.47 GHz for 45 mm-long actuators, and 2.43 GHz for 35 mm-long actuators.

The spacing between two inter-patterns also affects the heating performance as evaluated in Fig. 3E. The spacing between inter-patterns controls the width of the patterns, which slightly changes the resonant frequency, and the non-uniformity of the conductive traces, which controls the heating efficiency. It is shown that adequate spacing helps improve the speed of heating, as well as the resonant frequency. The resonant pattern reaches maximum resonance at 2.39, 2.40, and 2.41 GHz for inter-pattern spacing of 0.44, 0.47, and 0.5 mm, respectively.

The nozzle height configuration of the traces also affects the actuation result. During 3D printing, the nozzle deposits conductive ink onto the substrates. There is an inherent stochasticity in the process that creates non-uniformity and varying roughness on the conductive traces (Fig. 3C). The non-uniformity causes resistance changes. When current flows through the traces, the points where resistance is high increases current over that volume and creates more heat. Therefore, controlling the parameter of nozzle height during ink slurry deposition the non-uniformity and surface roughness of the traces and be controlled increasing the heating rate. With the printer nozzle set to 500 μm above the surface (compared to 200 μm for regular printing), the temperature increases to an average of 80 °C at 2.39 GHz, shown in Fig. 3F. Additional 2D plots of all frequency sweep tests can be found in the supplementary Section in Supplementary Figs. S3–5. For the pattern with a lower nozzle height, the resonant frequency appears at 2.40 GHz. However, as the nozzle is higher, the variation in temperature also rises since more uncertainty lies during the printing process.

For these actuators to reliably operate, they must not degrade either mechanically or electrically when receiving energy from the beamforming system over a high number of cycles. We conducted a cycle test that included over 500 actuation-cooling cycles powered by the beamforming system to validate the durability of the LCE actuator under wireless actuation and to find the breaking point of the actuator. The cycle test is conducted on three different samples, each tested more than 500 times, and Fig. 3G indicates the result of the cycle test of one actuator. We use a clay load of 20 grams. All 3 actuators are able to survive 500 cycles. Each heating process includes heating for 60 s by the beamforming system to ensure that the actuator arrives and maintains full actuation, and another 60 s to ensure that the actuator is cooled and relaxed. Until 548 cycles, the actuation and cooling speed, as well as the length of the actuator, do not change significantly.

The breaking point of the actuator happens on the conductive trace, where fracture and disconnection occurred after cycle 548. Figure 3I shows three different statuses of the conductive traces after cycles. In the second image, when the trace becomes damaged, EGaIn is released at the point of damage to fill and repair the gap. The self-heating procedure also causes some instability in the cycle test, as healing can take time to autonomously occur. However, during repeated cycles, the EGaIn begins to form more oxidation ($Ga_2O_3$) under heat and loses fluidity. Therefore, the trace finally yields and can not be further actuated. In such cases, since the mechanical property of the LCE has not degraded significantly, simply delaminating and removing the original traces and 3D-printing a new trace can re-establish its ability to operate.

## Beamforming-enabled locomotion of soft robots

RFact enables locomotion of a soft robot through frequency-dependent channel estimation and distributed beamforming. We leverage wireless beamforming at different frequencies to achieve the power concentration at the specific resonating frequency to the independent actuators. To accurately focus power on desired spots, we first discretely sample channel information through LED harvesters and a phase retrieval algorithm, and then use frequency-model interpolation and space-model interpolation to find out the channel information across the entire operation region and operating frequency. The wireless actuation process requires a collaborative signal transmission with 6 antennas and a total of 60 W maximum output power. To ensure safety and compatibility with FCC regulations, most experiments are done inside a Faraday cage. Our system is capable of selectively actuating a single actuator within 40 s, and at the same time, in parallel, an attached wireless power harvesting module is capable of providing a maximum of 400 mW harvested power from each actuator (in Supplementary Information Section 1 and Section 4). This demonstrates the potential of this system as both a transducer for thermal to mechanical actuation, along with electromagnetic to electrical conversion.

We present demonstrations showing different aspects of capabilities enabled by RFact. The frog and turtle imagery are added solely for demonstrating the jumping and crawling motion of a robot, and are not meant to suggest biologically mimicry of their natural locomotion. In the first example, we show the locomotion in the form of a motor-spring-latch system where beamforming is used to store elastic energy until a force threshold in the form of a magnet is met, followed by a fast release and high power output. In this case, a magnet-elastic band energy storage mechanism is used to store the mechanical energy obtained during LCE actuation. When the LCE is actuated to the maximum extent, the elastic energy is released in the form of kinetic energy to allow a jumping motion of the robot. The second example

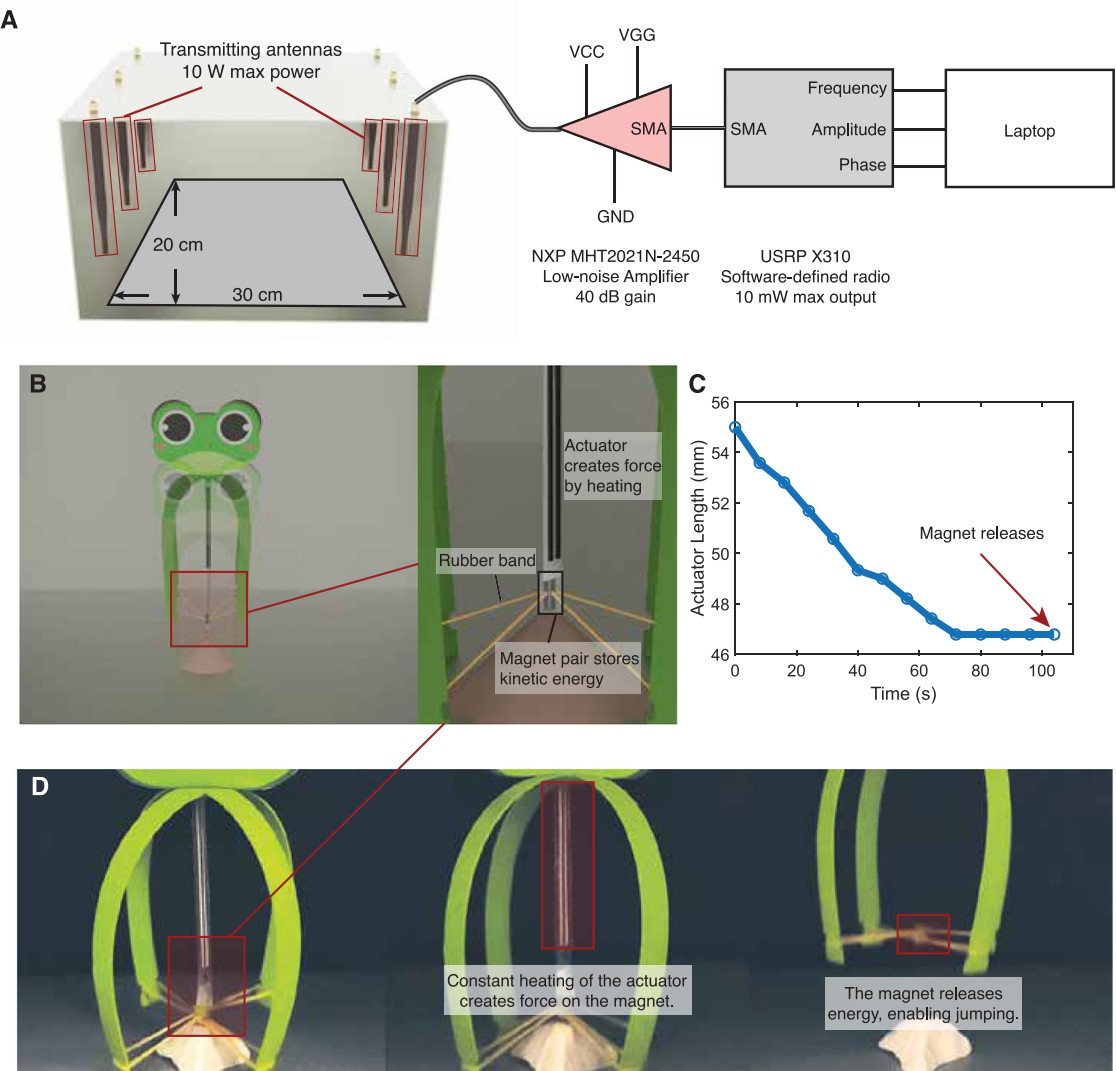

**Fig. 4 | Actuation setup and local actuation of a single actuator. A** The setup of RFact's frequency-selective beamforming system. **B** A jumping robot based on a motor-spring-latch system enabled by wireless actuation. The jumping robot is composed of an LCE actuator, an elastic rubber spring, and a magnetic latch for potential to kinetic energy storage and conversion. **C** The length of the actuator during the actuation period. **D** Screenshot of the actuation process. In the first step, the actuator is heated with wireless power and produces stress that is applied to the magnet. In the second step, the force is large enough to detach the magnet. When the magnet is detached, the stored energy pushes the jumping robot up.

demonstrates how frequency-aware selective actuation can be used to focus energy on different actuators separately. We demonstrate such capabilities through a crawling robot. With a controlled heating of the two actuators in three steps (heating left/right actuator and heating both actuators), the robot prototype can move bidirectionally along an axis. The third example shows a crawling soft robot operating inside a plastic pipe to pick an object. The robot is composed of two actuators, one to control movement and another to control lifting the object. The demonstration shows that the selective actuation scheme can efficiently operate robots in a fully-sealed environment through obstructions.

**Range of operation.** Figure 4A shows our system setup and dimensions of operation. This current implementation allows an operating range within a L × W × H = 30 cm × 20 cm × 20 cm cubic area, which is comparable to the operating areas of lasers[17,18] and NIR[20−23], but allows occlusions and accurate selective actuation of different actuators. The operating area is surrounded by six antennas, each with 10 W maximum transmitting power. The power of each antenna comes from a USRP software-defined radio (SDR) as the signal generator, further

passing through an NXP radio-frequency amplifier. We control the frequency, amplitude, and phase of the signals to achieve coherent beamforming by a laptop that is connected to the SDRs.

**Jumping robot with single actuator.** Our robotic platform is able to actuate actuators beyond obstructions. As shown in Supplementary Movie 4, an actuator that is hidden inside a 5 mm-thick plastic pipe can be actuated at a similar rate by our system with the same power output as the case without obstructions. To demonstrate our robotic platform, we further implemented a completely wireless and battery-free jumping motor-spring-latch robot. Inspired by ref. 42, the jumping robot is composed of an LCE actuator, a supporting structure made of plastic, two elastic bands for potential energy buildup, and a pair of 1/16"-wide 1/8"-long magnets (36 kJ/m³, 8300 G) to store potential energy. The structure of the jumping robot is shown in Fig. 4B. During the actuation process, wireless power is continuously beamed on the LCE actuator, heating and contracting it. The contraction generates a strain on the elastic bands for 90 s, reaching upwards of 120 °C until the force generated from the elastic potential balances the magnetic latching force. Figure 4C shows the length of the actuator decreases in

time from the initial 55 mm to 46.5 mm after a continuous wireless actuation of 70 s. After 70 s, the actuator length does not change, but the stress induced in the actuator still increases. After full actuation, the stress exceeds the maximum force produced by the magnets and releases the two magnets, enabling the jumping behavior. While the LCE actuator only weighs 1.5 grams, the jumping robot weighs 45 grams, with the height of jumping reaching 6 cm. Figure 4D shows the screenshot of three processes of the robot actuation, which are the initial state, during actuation, and magnet release. The demonstration indicates the ability of this system to bypass the lower power output of LCE actuation systems, which often need to generate a force over a long period of time, by instead storing the energy for a fast, higher power release, lifting an object 30 times the weight of the LCE actuator. A video is provided in Supplementary Movie 1.

**Walking robot with selective actuation.** Supplementary Movie 5 shows that through selective actuation and power delivery, we are able to selectively deliver power and information solely towards different actuators. We further show a demonstration of how wireless selective actuation can enable motion on soft robots with two actuators. Figure 5A–C and Supplementary Movie 2 present a walking robot with actuators inducing an antagonistic muscle alignment for movement. For this walker, two LCE actuators resonate at frequencies of 2.36 GHz and 2.4 GHz. By switching the wireless frequency between these two values, it is possible to move the leg upward when both actuators are heated, move down when both are cooled, and move left and right when only the left or right actuator is heated respectively, which is shown in Fig. 5A. Figure 5B further shows the length of the left and right actuator and the position of the foot during the whole actuation cycle, which corresponds to the screenshots in Fig. 5C. Initially, the leg is in the middle position at the beginning of the cycle. To move the robot right, we first actuate the left actuator, which will move the leg left by 1.5 cm over 30 seconds. The bottom of the leg is a foot made of sandpaper that creates friction with the ground. During the leg movement, since the foot does not move relative to the ground, the robot moves right. In the second step, we move the leg by increasing the actuation power of the right actuator and decreasing the actuation power of the left actuator for 30 s, which will make the leg move up and back to the initial position. We then wait for the leg to cool down for another round of actuation for 30 s. This produces an actuation cycle of 90 s for one gait with a step movement of 1.5 cm, which is 1/7 of the robot size.

**Crawling inside pipe.** A crawling robot demonstrates how RFact enables selective actuation under the plastic obstruction. The robot is operated in a fully-sealed plastic pipe with 5 mm thickness. Figure 5D–G and Supplementary Movie 6 show a crawling soft robot driven by the middle actuator. A plastic backbone provides structural stability for the robot and acts a restoring force after actuator cool down. Two copper feet are connected to the plastic backbone. They are cut into jigsaw patterns and bent towards one direction to provide directed anisotropic friction. Once the middle actuator actuates (30 s), it pulls the left foot towards the right by 1.7 cm. When it cools down and de-actuates (30 s), it pushes the right foot towards the right by another 1.7 cm. Through such cycles, the robot is able to crawl towards the right. Therefore, the cycle duration is 60 s for 3.4 cm of motion, which is 1/3 of the robot size. Another curling actuator is attached to the front of the soft robot. Curling is achieved by applying a layer of Sil-poxy on one side of the actuator, and when heated, the stiffness mismatch of the LCE and elastic Sil-poxy induces a curling along one direction. The tip of the curling actuator is covered with adhesive (Sil-poxy in this case). In the first step, we solely actuate the middle actuator, which controls crawling, to make the robot move into the position contacting the object to be picked. The curling actuator then adheres to the object after the adhesive cures. The motion of lifting the object can be achieved by actuating only the curling actuator. To move

the robot after picking the object, we simultaneously actuate the middle and curling actuator, so that the object does not touch the ground while the robot moves.

**Power harvesting during actuation.** In addition to the ability to be heated and drive the actuation, the resonant conductive trace can also behave as an antennas during actuation and support electrical power for on-board electronics. To demonstrate this capability, we connect an LED RF power harvester to the conductive trace on the actuator. Supplementary Movie 3 shows that the LED power harvester lights up when we apply more power to the actuator. The property of simultaneous actuation and power harvesting opens opportunities for the actuation system for more complex tasks that require the soft robot to carry on-board electronics. Additional information can be found in the Supplementary Information Section 8.

## Discussion
In this work, we introduce RFact, a radio-frequency actuation platform for LCE-based soft robots. RF stimulation enables power delivery through obstacles and occlusions, including through closed pipes, bypassing many of the line-of-sight limitations of NIR and laser light-based actuation approaches. RFact first introduces an LCE formulation with a low and tunable phase transition temperature as an effort to decrease the power requirements for RF-based actuation. Further, RFact introduces LCE and LM-based actuators and characterizes the heating behavior of the conductive traces with regard to frequency, for frequency-selective actuation. Finally, RFact proposes frequency-aware channel probing and interpolation to enable efficient beam-forming at different frequencies towards actuator locations.

To demonstrate the capability of RFact in the soft robotic field, we implement a series of robotic demonstrations. First, we show a jumping robot that demonstrates the ability of the LCE actuator to produce an impulsive force. Next, we present a walking robot that shows the ability to perform selective actuation in which individually addressable portions of the LCE can be stimulated. We then show actuation within a closed pipe and also demonstrate that RFact can be utilized for energy harvesting in its ability to power LEDs. Finally, we incorporate actuation inside an occlusion, demonstrating the system's ability to penetrate through structures and demonstrate a crawling robot actuating through a pipe, followed by picking up an object with a combination of a linear actuation and curling actuation.

### Future works
The demonstration presented here suggests that RFact could potentially be applicable in more practical and complex soft robotic use cases. As an example, RFact allows for power harvesting onboard the soft robot, which could enable a potential use case where the soft robot is embedded with sensors and control circuits and is capable of accomplishing more complex tasks without the need for an onboard power supply.

### Translatability
The concept and technologies of radio-frequency actuation proposed in RFact are translatable to other soft robotics systems. First, other thermal-responsive elastomers can be combined with resonant conductive patterns to be actuated by the actuation system proposed in RFact. Moreover, shape-memory alloys, if arranged in a specific resonant structure, can also be actuated using this approach. Second, beyond thermal-responsive actuators, the frequency-aware beam-forming system can be used with an antenna and energy harvesting board as a charging platform, which can be combined with additional types of electrically-responsive actuators. However, it is notable that our current measurement shows the electric power harvested from the actuator is far less than the thermal power, indicating that using electric power harvesters on our system may not create a comparably

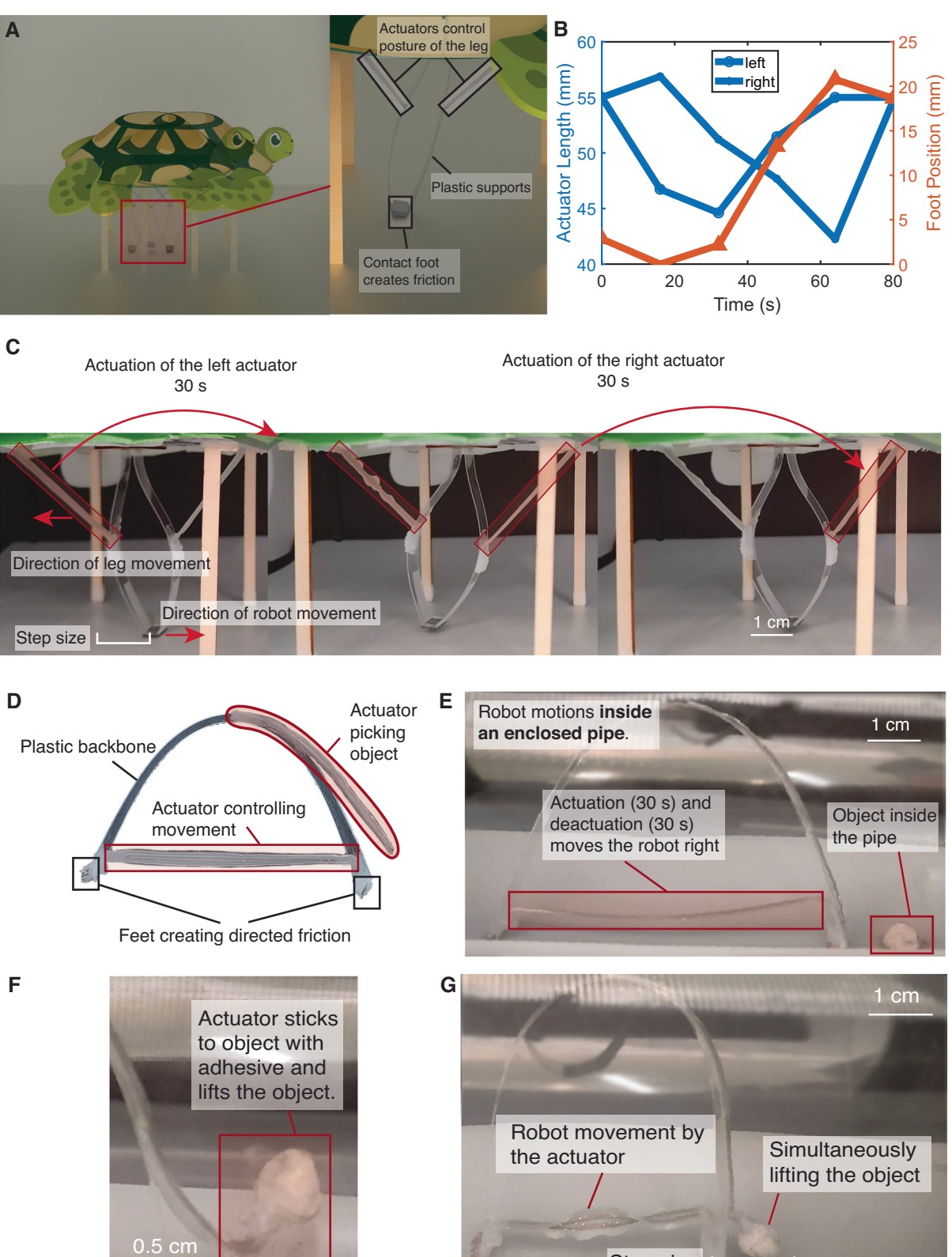

**Fig. 5 | Robotic demonstrations highlighting wireless selective actuation of soft robots. A** A walking robot enabled by selective actuation. The leg is composed of two actuators, a plastic support, and a foot to create friction. By solely actuating one actuator, the robot moves in the opposite direction. By simultaneously actuating both actuators, the leg returns to the center position. **B** The length of the left and right actuator, and the position of the leg in one plot, during one step of rightward motion. First, the left actuator is actuated, pulling the foot towards the left. Then, the right actuator is actuated, pulling the foot towards the left in the air. Finally, two actuators are released, while the whole robot moves right. **C** A screenshot of one gait for the robot to move right. **D** A soft robotic walker through a plastic occlusion. Two actuators are selectively activated with varying frequencies, enabling a robot to crawl inside a pipe and pick up an object. **E** Selectively actuating the central horizontal actuator can enable a crawling movement through aniso-tropically aligned friction from the copper feet. **F** Actuating the curling actuator on the front demonstrates lifting of the object. **G** Simultaneously actuating both actuators enables the robot movement while picking the object.

strong power output. Our platform can also be combined with battery-free wireless soft robots to achieve on-board charging. Finally, we propose an important concept of frequency-selective actuation. Although previous works on NIR light or laser actuation have demonstrated the frequency/wavelength resonance of certain actuators[43,44], RFact is capable of a more fine-grained (100 MHz separation) selective actuation scheme. The concept of frequency-selective actuation can be migrated to other soft robot platforms with other types of wireless power to enable wireless actuation of actuators that are arranged closely.

## Limitations

The current system uses a maximum of 60 W total power, which requires the deployment of a Faraday cage to comply with FCC and FDA regulations. This cage reduces the mobility of the system. Although it can be replaced by a flexible Faraday enclosure made with metallic sleeves, it still acts as an additional component of the system. The necessity of the Faraday cage acts as a protection for operators and is in compliance with the FCC rules, as we are operating in the Wi-Fi frequency. Therefore, the problem can be potentially solved by using an array of extra-directional antennas as transmitters, which do not point at operators, along with changing the operating frequency to an unlicensed band where not many transceivers exist. Second, the speed of LCE actuation and relaxation can potentially be improved. The slow speed arises from the time required for Joule heating and ambient cooling, which is a limitation common to other thermally-responsive actuators that use heat for stimulation[45]. Active cooling to speed up LCE relaxation[46] and a protective heat isolation layer to speed up LCE actuation are potential solutions for increasing performance. Finally, the failure of the conductive LM-Ag-SIS based elastic ink from oxidation and mechanical stress after high cyclical loading can not be avoided. We note that this limitation can be true of other flexible and stretchable electronics[47] and remains a challenge that remains to be fully addressed. Possible solutions include applying a thin protective layer of elastomer that leads to only a small increase in elastic stiffness without significantly affecting actuation.

## Methods

### Materials

The liquid crystal mesogen 2-Methyl-1,4-phenylene bis(4-(3-(acryloyloxy)propoxy)benzoate) (RM257) is purchased from ChemScene. 2,2′-(Ethylenedioxy)diethanethiol (EDDET), Pentaerythritol tetrakis(3-mercaptopropionate) (PETMP), Triethylamine (TEA), Bisphenol A ethoxylate diacrylate, and 2-Hydroxy-4′-(2-hydroxyethoxy)-2-methyl-propiophenone (HHMP) are purchased from Sigma-Aldrich. All chemicals are used without further purification.

### LCE Design

Regular LCE experiences a phase transition temperature varying between 70 to 90 °C[48]. Previous works have shown a decrement of LCE phase transition temperatures by using different monomer doping. For example, Bauman et al. used a monomer mixture of C6BAPE and C3M to decrease the phase transition temperature to near-ambient temperature of around 35 °C. However, the recipe is expensive since regular LCE costs around $1 per gram, and this recipe costs $80 per gram. We propose BisLCE that can be configured to different phase-transition temperatures by doping different proportions of Bisphenol-512 with monomer RM257. Our measurement shows that BisLCE can be configured to actuate from ambient temperature (22 °C) to 75 °C regarding other soft-actuator use cases. BisLCE is also more accessible in cost as it shares a similar cost as regular LCE. A tradeoff is done to determine the actuation temperature of the LCE, where lower $T_{ni}$ leads to a lower power requirement but a better resolution for selective actuation asks for a higher $T_{ni}$. Therefore, we synthesize the LCE to have a phase-transition temperature at 53 °C.

### Fabrication methods

To fabricate the LCE, molds with a size of 25 mm × 50 mm × 1 mm are adopted. The fabrication procedure is as follows. Firstly, 4.500 g of the liquid crystal RM257, and 0.0054 g of the photoinitiator HHMP, 0.206 g of Bisphenol A ethoxylate diacrylate ($M_n$ ~ 512), and 1.468 g of toluene are added in turn to a glass vial and heated up to 80 °C with a heat gun until the powders are fully dissolved. After cooling to room temperature, 1.058 ml of EDDET, 0.219 ml of PETMP, and 0.639 ml of TEA are added to the vial with a micropipette. The mixture is then mixed at 2000 rmp for 1 min in a planetary mixer. After mixing, the mixture is poured into the molds and cured for 12 h. After 12 h, LCE should have cured in the molds. They are demolded and put in a vacuum oven to remove the toluene solvent at 80 °C and 508 mmHg for 12 h. Finally, the fabricated LCE is stretched to a strain of 1.5 along the 50 mm direction, then cured in a UV cross-linker for 3 min for both sides, and then they are cut into the width of 3 mm. Supplementary Information Section 5 provides a table that describes the recipe of LCE with different proportion configurations of Bisphenol-512.

### Differential scanning calorimetry

Differential Scanning Calorimetry is conducted on a differential scanning calorimeter (Q20; TA Instruments). One sample of approximately 10 mg is prepared by first stretching a strip of LCE and then using UV light to activate a UV crosslinker (CL-1000 Ultraviolet Crosslinker; UVP) for 6 min. The samples are conditioned at room temperature for 30 min. Samples are heated from room temperature to 120 °C, cooled to −30 °C, then heated to 120 °C. Heating and cooling steps are performed at 10 °C/min and 5 °C/min, respectively. The heat flow-temperature curves from the second heating step are used to determine the phase-transition temperature.

### Dynamic mechanical analysis

Dynamic Mechanical Analysis is conducted using a solid analyzer (RSA-G2; TA Instrument). The temperature is ramped from 20 °C to 100 °C at a rate of 3 °C/ min with uniaxial load controlled at 0 N.

### Force-temperature and stress-temperature test

The force-temperature and stress-temperature tests are conducted using a materials testing machine (Model 5969; Instron). The strain is maintained at 0% during the test, while the temperature rises linearly in 30 s from 0 °C to 100 °C.

### Power density and work density test

The power density and work density tests are conducted using a materials testing machine (Model 5969; Instron). The load is maintained at 0.5 N during the test, while temperature rises linearly in 30 s from 0 °C to 100 °C.

### Actuator packaging

To accomplish this and enable stretchable and electro-mechanically stable conductive traces, eutectic gallium indium (EGaIn) liquid metal, silver flakes, and styrene-isoprene-styrene block copolymers (SIS) were combined to make a 3D printable conductive ink (LM-Ag-SIS ink) to serve as the conductive antenna. The conductive ink is made by mixing these materials following the steps described in ref. 40, which is adapted from LM-Ag-SIS ink formulations introduced in refs. 41,49. Once printed with direct ink writing (DIW), the solvent evaporates from the printed ink and the traces become electrically conductive. This process begins with the first LCE layer oligomerized. This is followed by applying a uniaxial strain and photo-polymerization cross-linking to fix the actuation direction by freezing the molecular organization. The actuator is made from one piece of LCE with the printed conductive traces embedded on top of it. Two parallel LM-Ag-SIS ink traces are then DIW printed over this layer to act as the frequency-dependent RF absorber.

This fabrication process enables rapid manufacturing of actuators with varying antenna dimensions. To provide better selectivity and induce more complex multi-actuator gaits, we selectively control the heating of different LCE actuators by tuning the dimensions and trace width separations of these LM-Ag-SIS traces to resonate at different frequencies. This allows for multi-actuator arrays of these soft robotic muscles to be integrated near each other while still being individually addressable. This is done by tuning the frequencies of the output antennas to resonate with various-sized LCE actuators that resonate at that frequency. This is done to ensure wireless-controlled actuation of individual actuators in a multi-actuator array by controlling frequency and location of beamforming. This system can also operate without the need for line of sight that NIR and optical wavelength wireless energy transfer[20,50,51] require, allowing the system to operate while in tight, hard-to-reach environments.

## Beamforming-based wireless actuation

The LCE-based actuators are capable of being actuated by RF waves with frequencies ranging between 2.35 GHz and 2.5 GHz. To further improve the actuation efficiency and selective actuation accuracy, a wireless beamforming system is designed and built. Six RF chains (10 W each) in the beamforming system are coordinated by a laptop to control the power distribution over the area of interest. In this work, a Faraday cage is placed surrounding the test environment to prevent leakage, in compliance with FCC and FDA regulations. However, with more directional antennas and RF chains, the system has the potential to be extended over the open space.

To actuate this material architecture, we designed and built a test setup for the wireless actuation of the soft robot. Regular beamforming requires the receiver to provide channel feedback[52]. However, LCE actuators are not connected to any active electronics and thus could not provide channel feedback. To compensate, we deploy several 2.4 GHz LED RF energy harvesters around the area of actuation. We could infer the received power of the LED harvesters according to their brightness. By changing the beamforming factors at the antennas, the brightness of LED harvesters changes, and therefore, we could infer the channel from the transmitters to the LED harvesters through the brightness change. To realize this, we adopt a phase retrieval algorithm to calculate the channels at the location where the LED harvesters are located. The channels at arbitrary locations can be further inferred by interpolating the probed values. During the characterization process, we fixed the known position of the LED actuators and set corresponding beamforming vectors at the antennas, such that the maximum amount of energy is always accessible to the actuators being tested.

We use a 60 W microwave setup to actuate the LCE actuators by beamforming. The operating frequency is selected between 2.35 GHz and 2.5 GHz due to the following reasons. First, to enable non-line-of-sight actuation, the operating frequency can not be too high. Sub-6 GHz EM waves still preserve the capability to penetrate through common blockages such as plastic, wood, and bricks without losing too much energy[53]. In contrast, high-frequency signals such as lasers, NIR lights, and millimeter waves are even hardly able to penetrate through papers[54]. We select relatively high frequencies so that we could have a smaller wavelength, and thus we could have a smaller beamforming area and higher power efficiency and heating accuracy towards different actuators located in different places.

We develop a wireless beamforming technique inside the EM enclosure to focus wireless energy on the actuators. To enable beamforming, we first probe the wireless channel around the soft robot with LED energy harvesters with a 2.4 GHz coil antenna. The LED energy harvesters provide brightness feedback according to the EM power level it perceives. We implement a PhaseLift[55] algorithm in MATLAB to infer the channels at the locations of the LEDs according to the probed brightness levels. The algorithm works as follows. Given a

set of $N$ probes with complex weight $\mathbf{W}_1, \mathbf{W}_2, \cdots, \mathbf{W}_N$, the received power of each LED $k$ with channel $\mathbf{h}_k$ can be described as $p_{n,k} = |\mathbf{W}'_n \mathbf{h}_k|$. Therefore, to solve the channel $\mathbf{h}_k$, we solve the problem

$$\min_{\mathbf{h}_k} \quad \| \, |\mathbf{W}^\top \mathbf{h}_k| - \mathbf{p}_k \|_2^2 \tag{1}$$

The problem is solved using a MATLAB implementation of the PhaseLift algorithm.

After inferring the channel values at different LED harvester locations, the channel values across the whole plane where the LEDs are distributed can be obtained by the method of interpolation. The interpolation is based on the fact that the EM waves propagating inside the Faraday enclosure form standing waves due to reflective boundaries. The standing wave formula is dependent on the geometry of the Faraday enclosure and the operating frequency. According to the channel model, we can find the channel values at arbitrary places by the following rule:

- The amplitude of the channel value is interpolated in space according to the square of a sinusoidal function. This is due to the standing wave formula inside a closed space being expressed by a square of a sinusoid.
- The phase of the channel value is interpolated in space according to an affine function. This is due to the phase of standing waves in space changing linearly.
- The amplitude of the channel value is interpolated in frequency according to a sinusoidal function. This is due to the nodes of a standing wave moving linearly with the frequency.
- The phase of the channel value is interpolated in frequency according to aa affine function. This is due to the phase of the standing wave is inversely proportional to the wavelength, and thus linearly proportional to the frequency.

We provide detailed formulas in Supplementary Information Section 6.

Beamforming is performed on top of the probed channel values. We set the phase of the signal radiated by each antenna to be the inverse value of the phase of the probed channel, so that the EM waves from all the antennas add up coherently at the desired heating point. Given our channel measurement of $\mathbf{h}$, the beamforming vector $\mathbf{v}$ with phasors $\mathbf{v}_1, \mathbf{v}_2, \cdots, \mathbf{v}_N$ representing the transmitted EM waves from each antenna can be inferred as

$$\mathbf{v}_n = P_n \frac{\mathbf{h}'_n}{|\mathbf{h}_n|} \tag{2}$$

## Hardware and software implementation

We use 3 USRP X310 SDRs, each equipped with 2 RF chains as the signal source. The SDRs are connected to a laptop with 24 GB memory and a 10 Gbps ethernet port through a NetGear GS105 switch and Cat5 ethernet cables. The signal transmitted by the SDRs is amplified by an NXP MHT2021N-2450 linear power amplifier that has a maximum 42 dB amplification ratio. The amplifiers are powered by DC sources with 5.29 V VGG and 28.0 V VCC. The Faraday cage we use is a modified Sharp SMC1441CW microwave oven without the wave port chamber. Six omnidirectional 2.45 GHz antennas (VERT2450) are placed at locations ($-13/0/13$ cm, $-16/16$ cm), with the center of the chamber as (0 cm,0 cm). The operation region of the robot is bounded by the area $x \in (-10, 10)$ cm, $y \in (-15, 15)$ cm.

For channel estimation, we deploy an array of LED harvesters (Schematic and layout in Supplementary Information Section 3) with 1 cm × 1 cm separation. We use 24 random probes from 6 antennas and record the brightness of each LED energy harvester by counting the bright pixel numbers captured by an ELP 4 mm Lens Prototype Camera to get the spatial distribution of a single frequency. We probe the

channel in 2.35, 2.36, ⋯ , 2.49, 2.50 GHz, and then interpolate the rest spatial and frequency channels by the previous channel inference method. The method is implemented in MATLAB and run offline.

## Frequency test methods
During the test of resonating frequencies of actuators, we select a beamforming vector for each antenna that results in a similar power level for the actuator in all frequencies from 2.35 GHz to 2.5 GHz. This is done by utilizing the beamforming formulation described in Eq. (2). By probing the channel vectors $\mathbf{h}$ in different frequencies, we know that the output power of the optimal beamforming vector is $P_n\|\mathbf{h}\|_2$. Therefore, we scale the beamforming vector down by $1/\|\mathbf{h}\|_2$ such that the overall received power at the actuator would stay at $P_n$, which is the power transmitted at a single antenna. Taking $P_n$ to the maximum 10 W output would result in quick burning of actuators in resonating frequencies under our 30 s heating setup. Therefore, we tune the transmitting power down by 6 dB, which results in a total receiving power of around 2.5 W for a single actuator.

## Robot actuation
The robot is actuated in the open loop, where the beamforming vector is pre-calculated according to the location of the robot. For the jumping demo, we use a constant beamforming vector given by Eq. (2), where each antenna emits a full 10 W power level. For the power harvesting demo, the power at each antenna grows logarithmically (the dB value grows linearly) with time from 0 dBm (1 mW) to 40 dBm (10 W) for each antenna, and the phase of the beamforming vector is acquired through Eq. (2). For the crawling demo, we calculate the beamforming vector along the moving trace of the robot. The 3 actuation steps are decoupled, and the beamforming vector is pre-determined through the position of the moving robot calculated through Eq. (2). Through tracking techniques such as millimeter-wave radar or cameras, it would be possible for robots with wireless actuation to form a closed-loop control of the robots, given the current status and position of the robot.

## Data availability
All the recipes, codes, processed results, and raw results (DSC/Mechanical tests, RF tests, channel measurement results) will be shared in the public repository (https://doi.org/10.6084/m9.figshare.27987485). Due to the large file size, some raw measurements (raw channel measurement signals, temperature maps) are available upon request.

## Code availability
Code for the beamforming system are shared in the public repository (https://doi.org/10.6084/m9.figshare.27987485), including the channel estimation and beamforming with software-defined radios.

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

## Acknowledgements

We thank the support from NSF (2030154, 2106921, 2007786, 1942902, 2111751), ONR, DARPA-SOAP, CyLab-Enterprise, and AFRL Center for Excellence for Data-Driven Design of Multifunctional Material Systems (D3OM2S).

## Author contributions

Y.S., Z.L., M.Z., J.W., S.K., and C.M. produced the idea together and did the revision of the manuscript. Y.S., Z.L., and M.Z. conducted material fabrication and material characterization. Y.S. and Z.L. conducted radio-frequency experiments. Y.S. and Z.L. wrote the original manuscript. M.Z., J.W., S.K., and C.M. wrote and edited the manuscript. S.K. and C.M. collected fundings for the project.

## Competing interests

The authors declare no competing interests.

## Additional information

**Peer review information** *Nature Communications* thanks Mihai Duduta, Chongjing Cao, and the other anonymous reviewer(s) for their con-tribution to the peer review of this work. A peer review file is available.

