## [Transparent Peer Review file · Nature Communications]

Frequency-Selective Actuation of Liquid Crystalline Elastomer Actuators with Radio-Frequency

Corresponding Author: Mr Yiwen Song

Version 0:

Reviewer comments:

Reviewer #1

(Remarks to the Author)

This paper proposes the use of GHz-level radio frequency signals to drive soft robots. Undoubtedly, wireless radio frequency technology is crucial for the development of wireless soft robots, especially those that need to operate in narrow, enclosed spaces such as the human gastrointestinal tract. However, it is unfortunate that the authors did not showcase an outstanding work based on this technology. Although the authors demonstrated that wireless radio frequency technology can drive LCE deformation, the design of the soft robot is too simplistic, resulting in insufficient innovation in the paper. Despite the authors' team being highly professional in the field of robotics, I regret to say that this paper does not meet the standards of Nature Communications and falls short compared to similar articles. Here, I must regretfully provide some reasons and opinions regarding the design and innovation of the paper.

1. The authors stated that to address the challenge of driving soft robots in narrow or confined environments, they proposed the use of radio frequency power supply technology. However, the two soft robot demos presented by the authors have no connection to narrow or enclosed spaces. In other words, the proposed technology was not used or demonstrated in the appropriate context.
2. This paper is an interdisciplinary fusion of technologies, with each technology not being innovative in its respective field. For example, radio frequency power supply technology is a fundamental technology in the fields of electronics and communications, and the development of low-temperature LCE materials is also a basic LCE formulation. Therefore, the innovation of this paper depends on the effectiveness of the application of these technologies. However, based on the demonstrations provided by the authors, the applicability is very weak.
3. The two soft robots developed by the authors are designed with cartoonish frog and turtle appearances. However, it must be said that these two robots do not mimic the movements of frogs and turtles. Frogs jump using their four legs, especially their two hind legs, which is not similar to the soft robot demonstrated by the authors. The authors only showcased a bouncing design, which is not capable of repeated bouncing. Turtles crawl using their four soft legs, but the authors used four wooden sticks to support a turtle-shaped design, which is also quite odd.
4. The title of this paper is too broad and does not highlight the technical features of the article. Additionally, radio frequency-driven soft robots have already been adopted by many researchers.
5. The third example presented by the authors involves using an LCE actuator to harvest wireless energy to light up an LED. This demo seems unrelated to soft robots. It merely demonstrates that wireless power can light up an LED, which has no connection to the deformation and movement of robots.

(Remarks on code availability)

Reviewer #2

(Remarks to the Author)

The authors follow up on a couple of conference proceedings to describe a more detailed version of a more complex system, capable of RF power delivery to a soft robot. Overall the work is timely, well supported by relevant references, and a significant advancement on the field. This reviewer also appreciates the transparency on disclosing the past conference proceedings and clear delineation on why this is a separate and distinct body of work. Some suggestions for improvement are listed, before the article almost ready for publication, will be ready for a broader audience.

1. The authors claim this work uses a better actuator, but don't provide a head to head comparison of the actuators, nor a clear benchmarking of the current elastomer. For example, what's the work density of this actuator? What about the power density? How do they compare with other LCEs, or SMAs, or other soft actuators. Even if they don't win on all the metrics, it's still important to the field to show the trade-offs, since this is the one system that is both soft and RF powered.

2. The system required to drive this seems somewhat complex. A quick search of the power unit showed it's roughly \$10k which would limit adoption. Still, what would be most interesting to this reviewer is a photo at scale showing how much hardware is needed to drive how much actuator. Often in soft robotics some of these details are hidden from the reader, which obscures the message. In the interest of transparency, please add a photo of the entire system required to deliver RF power as described in section 4.8.

3. Do the authors consider the test in Figure 3G to be sufficient to show the actuators are "reliable"? Thinking about possible applications, like the autonomous robot described, that would correspond to about 100 steps. While on this topic, can Figure 4 and the text around it give a clear indication of the speed of the robot in body lengths per second, as is the standard in robotic locomotion demonstrations? Even if slow, it needs to be disclosed for direct comparisons. Coming back to the 100 actuation cycles, this seems insufficient for a robotic application. Can the authors propose an application where 100 cycles would be sufficient? Additionally, the way data is shown in Figure 3G is hard to read, because it's compressed on the vertical axis. What about also showing in the supplemental information a figure where the total contraction is plotted as a fraction of the initial contraction, and have the scale be say 95% - 105%, that would show the reader the variability in actuation much more closely.

4. The concept is intriguing and likely to make a significant impact on soft robotics. Can the authors comment on how translatable would this RF power delivery be for other soft robotics systems, beyond LCEs?

(Remarks on code availability)

N/A

Reviewer #3

(Remarks to the Author)

In this manuscript, Song and Li et al. developed a type of radio frequency actuated LCE for soft robotics. The material and characterizations of the LCE were introduced and applications of the RFact were demonstrated. The reviewer believes that the significance and contribution of this paper are below the baseline for publication in NC in its current form. Therefore, at least substantial revisions are required before this paper can be considered further for publication.

One of the major comments from the reviewer is the contradictions between these promising statements of the RFact in the Introduction and the weak performances in the demonstrations in later sections. In the Introduction, the authors summarized all the limitations of other soft actuators in mobile, untethered soft robotics and therefore claimed that their RFact can 'enable more complex movements of soft robots', 'enable a series of lightweight soft robotics that can navigate through their environment'. If these can be properly achieved in this paper, the reviewer believes that the significance and contribution of this paper will be sufficient for publication. However, the current demonstration applications in this paper, such as the weak jumping of the frog or the leg motion of the turtle, are far from 'complex movements' and 'navigate through environment' that the reviewer expects. The reviewer suggests the authors to develop proper soft robotic applications based on their RFact that demonstrate substantial scientific and technological advancements over the state-of-the-art.

The authors are suggested to summarize the major contributions more clearly in the Introduction. The reviewer noticed that the RFact developed in this paper was extended from many previous works by the authors, as the authors accurately stated in the Introduction and in Results page 4. However, this makes the contributions of this paper unclear, i.e. it reads like this is a paper that simply mixes some ingredients from the authors' different previous works. The authors are suggested to reorganize the Introduction which clearly states the differences between this paper and all related previous works, the challenges solved when adopting these previous works for this paper, and the major contributions of this paper.

The demonstrations in the current version of this paper could benefit from clearer explanations and illustrations of their working principles in Figure 4. For instance, the mechanical design of the frog can be plotted. How is the energy stored, how is this energy released can be drawn more clearly. In the current form, these three demonstrations are restricted in such a small figure, which is completely unnecessary.

Some minor comments:

1. The last paragraph in Introduction, why this paragraph should exist? Is it for introducing the structure of this paper?
2. Figure 1, (A) to (I) captions are wrongly located.
3. The reviewer suggests the authors to enrich the information in all their figures. Please seek suggestions from the senior authors of this paper.
4. In page 4, the authors stated that the large-scale shape change makes LCE an excellent candidate material for actuators in soft robotics. Please comment on the response rate and actuation period of the LCEs as well.
5. Figure 2, the quality of C to E should be improved. The font sizes in these four data plots should be consistent. The unit for temperature °C looks different in D. The reviewer also suggests to enlarge these data plots, the small sizes make seeking information from these plots difficult.
6. Page 6, the LCE actuator was tested in cycles, please provide information on how many cycles were tested, either in the text or in the figure.

7. Page 6 bottom, the authors directed the images of the completed actuator to Figure 1A. However, there isn't much information regarding the details of this actuator. More information exists in Figure 3A, B instead.

8. Figure 3 D to F. It would be beneficial to provide the frequency at which temperature peaks occur. This is related to the reviewer's comment on enriching the information in these figures.

(Remarks on code availability)

Version 1:

Reviewer comments:

Reviewer #1

(Remarks to the Author)

The author's revisions did not fundamentally enhance the novelty of the article. From a technical perspective, I find the paper feasible and have no objections in that regard. My concerns about the article's innovation primarily stem from the following two aspects:

First, in the field of soft robotics, published works on soft robots consistently demonstrate rich and robust experimental demonstrations. In this regard, the gap between the authors' work and existing literature remains significant. Furthermore, I believe the cartoon depictions of the frog and turtle undermine the article's persuasiveness and demonstrative impact.

Second, the Figures in the article cannot show clear innovation. Is the novelty centered on the low-temperature phase-change liquid crystal material, the radio-frequency (RF) wireless powering method, or the frequency-selective RF powering? Alternatively, does the system's functionality achieve something that cannot be done by other methods? If the latter is the case, the experimental demonstration presented by the author does not seem to be enough good. The low-temperature phase-change material, in my view, is not novel and is widely utilized in liquid crystal research. Regarding the frequency-selective RF powering and wireless powering methods, while the authors have published related studies, these aspects remain underexplored. However, the authors have failed to provide sufficient new data, demonstrations, or detailed analyses to substantiate deeper application-level advancements.

(Remarks on code availability)

Reviewer #2

(Remarks to the Author)

Overall the reviewers are in agreement that at its start the paper did not meet the high standards of this journal. After seeing the level of work that went into the revised work, I'm pleased to say that the new version of the article is more clear:

-it clearly delineates differences between this work and the preceding literature from the same research group

-it shows a more reliable actuator, up to 500 cycles from the original 100

-the power and work density characterization is more thorough, and shows the advantages of this technology.

I consider the work improved sufficiently to be acceptable for publication, thanks for giving me the opportunity to review.

(Remarks on code availability)

N/A

Reviewer #3

(Remarks to the Author)

My comments have been addressed. Well done!

(Remarks on code availability)

My comments have been addressed. No further comments from this reviewer.

RESPONSE TO REVIEWERS

We gratefully thank the reviewers for their efforts in reviewing the paper and their insightful comments. In the revised manuscript, we have made an effort to improve the paper. We summarize our major changes as follows.

First, we have added more experiments on multiple aspects, including LCE characterization, actuator cycle tests, and demos.

- (1) We show a comparison between our LCE with 5% Bisphenol-512, regular LCE with 0% Bisphenol-512, and our LCE after 500 cycles to demonstrate a difference in the mechanical properties. The changes are reflected in Figure 2.
- (2) We show a cycle test with significantly more cycles (over 500 cycles) until the breaking point of the actuator. We also analyze and discuss the reason of the breaking point.
- (3) We add an extra demonstration for the use case of our actuation platform, showing a crawling robot operating inside of a pipe and picking up an object.

Second, we revised the writing of the paper to emphasize our contribution to the design of the actuator and the actuation system, and show the differences between our study and existing works. In summary, past works focus on studying the feasibility of microwave heating of the elastomers and efficiently delivering microwave heating through beamforming. This work introduces a first proof of concept prototype of a self-sustaining actuation system towards a wireless actuation platform that achieves comparable range of mobility as soft robots with on-board batteries – e.g. crawling, hopping, picking up objects, and power harvesting to power electronics.

Third, we add in the discussion section remarks on limitations and future work based on our system. In short, the two major limitations of our system is that the requirement of using a Faraday enclosure restricts mobility, and that the de-actuation process of actuators is slow due to passive cooling. This work opens up spaces for future research on frequency-division power delivery for soft robots and other robotic systems, actuator designs, and frequency-aware RF responsive patterns.

Appended is our revised manuscript, where the blue text shows the difference between the original manuscript and the revised manuscript.

The following carefully describes the changes we have made in the revised manuscript:

[Response to R1]

This paper proposes the use of GHz-level radio frequency signals to drive soft robots. Undoubtedly, wireless radio frequency technology is crucial for the development of wireless soft robots, especially those that need to operate in narrow, enclosed spaces such as the human gastrointestinal tract. However, it is unfortunate that the authors did not showcase an outstanding work based on this technology. Although the authors

demonstrated that wireless radio frequency technology can drive LCE deformation, the design of the soft robot is too simplistic, resulting in insufficient innovation in the paper. Despite the authors' team being highly professional in the field of robotics, I regret to say that this paper does not meet the standards of Nature Communications and falls short compared to similar articles. Here, I must regretfully provide some reasons and opinions regarding the design and innovation of the paper.

[1] The authors stated that to address the challenge of driving soft robots in narrow or confined environments, they proposed the use of radio frequency power supply technology. However, the two soft robot demos presented by the authors have no connection to narrow or enclosed spaces. In other words, the proposed technology was not used or demonstrated in the appropriate context.

We thank the reviewer's advice to strengthen the demos with robots working inside narrow or enclosed spaces. To address the concern, we have added a new demonstration, shown in Supplementary Movie S6. In the demo, we show a working robot crawling and picking up an object inside an enclosed plastic pipe of 5 mm thickness. The robot has two actuators, and our system is capable of selectively actuating each actuator to accomplish moving and lifting the object. To describe the demo, we have added a new Figure 4D and corresponding text revisions in Lines 319-336 on Page 12-13.

Figure 4D: Wireless actuation enables a robot to crawl inside a pipe and pick up an object. Selectively actuating the middle actuator can enable crawling movement through the directed friction from the copper feet. Actuating the curling actuator enables lifting of the object.

Crawling inside pipe: A crawling robot demonstrates how RFact enables selective actuation under the plastic obstruction. The robot is operated in a fully-sealed plastic pipe with 5 mm thickness. Supplementary Movie 6 shows a crawling soft robot driven by the middle actuator. A plastic backbone provides structural stability for the robot. Two copper feet are connected to the plastic backbone. They are cut into jigsaw patterns and bent towards one direction to provide directed friction. Once the middle actuator actuates (30 seconds), it pulls the left foot towards the right by 1.7 cm. When it cools down and relaxes (30 seconds), it pushes the right foot towards outwards by another 1.7 cm. Through such cycles, the robot is able to crawl towards the right. Therefore, the cycle duration is 60 seconds for 3.4 cm of motion, which is 1/3 of the robot size. Another curling actuator is attached to the front of the soft robot. Curling is achieved by applying a layer of Sil-poxy on one side of the actuator, and when heated, the actuator will curl towards the other side. The tip of the curling actuator is covered with adhesive (Sil-poxy in our case). In the first step, we solely actuate the middle actuator, which controls crawling, to make

the robot contact the object to be picked. The curling actuator then adheres to the object after the adhesive cures. The motion of lifting the object can be achieved by actuating only the curling actuator. To move the robot after picking the object, we simultaneously actuate the middle and curling actuator, so that the object does not touch the ground while the robot moves.

[2] This paper is an interdisciplinary fusion of technologies, with each technology not being innovative in its respective field. For example, radio frequency power supply technology is a fundamental technology in the fields of electronics and communications, and the development of low-temperature LCE materials is also a basic LCE formulation. Therefore, the innovation of this paper depends on the effectiveness of the application of these technologies. However, based on the demonstrations provided by the authors, the applicability is very weak.

We thank the reviewer for their feedback. We acknowledge that in our previous description, the technical challenges and novelties of our paper may not have been fully clarified. We revised the introduction and results section to address these concerns. Specifically we highlight how this system addresses the fundamental challenges and limitations that NIR, laser, and magnetic based actuation approaches currently have along with highlighting the individual chemistry, materials, and RF improvements here. We summarize the contribution of our system as follows in Lines 73-75 of the Introduction on Page 2:

RFact serves as the first proof-of-concept system towards a wireless actuation platform that achieves a comparable range of mobility as soft robots with on-board batteries [31, 32] – e.g. crawling, hopping, picking up objects, and power harvesting to power electronics.

The main contribution of our approach is that we achieve efficient and selective RF-based actuation of LCE actuators. While our previous work demonstrated microwave heating of liquid metal patch antennas to introduce RF based actuation, this work introduces a proof of concept RF soft robotic system addressing the direct limitations of NIR, laser, and magnetic near field wireless technologies. Here, we divide the novelty of our system into three parts.

First, on the low-temperature LCE side, we study a simple method by mixing Bisphenol-512 into RM257 to create LCE with lower phase transition temperatures. Compared to past methods [1*,2*], the method we propose is a cheaper yet effective method. Moreover, we can control the thermo-response temperature between body temperature and 74°C, depending on the fraction of Bisphenol-512. Most importantly, according to the mechanical analysis, the LCE with 5% Bisphenol-512 achieves lower phase transition temperature, while providing similar force output, stress output, and power/work density when compared to LCE with 0% Bisphenol-512.

Second, we study a resonance conductive pattern that can be 3D printed on the LCE to achieve frequency-selective heating. Past work on resonating patterns for antennas and backscatter mainly focuses on how to deliver the RF power to other circuit components. In contrast, we

study how to efficiently convert RF power to heat with the 3D printed patterns, while maintaining the 3D printed patterns to be in a very narrow-band to achieve selective actuation.

Finally, we study a frequency-dependent beamforming algorithm. RF power supplies for electronics and communications require a channel feedback from the recipient, which does not exist in our case. Therefore, we use passive beamforming with a channel probing in advance. Past adaptive beamforming assumes the same channel frequency, whereas in this paper we study the possibility of beamforming with single probing but beamforming at different frequencies.

[1*] R. K. Shaha, A. H. Torbati, and C. P. Frick, "Body-temperature shape-shifting liquid crystal elastomers," *Journal of Applied Polymer Science*, vol. 138, no. 14, p. 50136, 2021.

[2*] G. E. Bauman, J. M. McCracken, and T. J. White, "Actuation of liquid crystalline elastomers at or below ambient temperature," *Angewandte Chemie International Edition*, 2022.

The three changes are reflected in the revised Introduction in Lines 60-72 on Page 2:

In this paper, we introduce RFact, a soft robotics platform to achieve accurate, electronics-free wireless actuation that is battery-free, achieving long-range operation to 30 centimeters, the capability to operate through obstructions, and the ability to selectively actuate different actuators. RFact achieves this through a combined design of LCE chemistry, soft actuator integration design, and a radio-frequency beamforming system. First, RFact introduces a recipe for an LCE actuator that responds to relatively low temperature stimulation to fit the context of wireless actuation with less power. Second, RFact exploits the frequency-selective actuation capability of the LCE-based actuators. This is achieved by integrating the actuators with 3D-printed conductive traces that are tuned to resonate at different frequencies. The soft actuator design allows RF heating as well as harvesting power for low-power electronics. Third, RFact proposes a frequency-aware channel interpolation algorithm for beamforming towards a desired actuator location. The frequency-aware beamforming allows actuation at different places and different frequencies through measuring the channel at a few locations in advance.

Moreover, at the beginning of the Results section, we expand more to describe the contributions and novelties of our paper in Lines 80-103 on Page 4:

Thermally responsive LCEs typically have phase transition temperatures above 70 °C [32], which requires a large amount of power to achieve full actuation. To actuate LCE actuators towards such a high temperature, high RF input power is used to heat a small region with a small actuator. We aim to activate the LCE actuators with a similar amount of power used in our previous work but with LCE actuators that are of larger size over a larger area. To achieve this, RFact makes use of LCEs that are tuned to have a lower phase transition temperature. The tunable phase transition temperature allows users to choose between high actuation selectivity (co-operate with frequency-selective actuation) and low-power actuation. In particular, we present a method to selectively mix monomers to weaken the interaction between chain bonds by looking into the chemical mechanism of LCE actuation. Specifically we use mixed monomers

to weaken the $\pi - \pi$ interaction in the LCE at the nematic phase. Second, we study the resonance of 3D-printed conductive traces. Our past works have shown that using metal coating or conductive-trace coating on LCE can significantly improve actuation speed due to induction heating. This work conducts an extensive study on how the design and fabrication of conductive traces would affect the resonant frequency and heating speed. We then package the LCE with the 3D-printed conductive traces to fabricate actuators that are frequency-selective. Frequency-selective actuators in close spatial regions can be selectively actuated.

Finally, we study the frequency-aware beamforming. Beamforming is achieved through a network of multiple antennas that coherently combines power towards desired locations. The frequency-selective actuation creates new challenges for beamforming since the wireless channel changes with frequencies. We study the standing waves inside the environment and use frequency-aware interpolation to create beamforming weights corresponding to the current operating frequency and the location to be beamformed. The method allows few-shot probes of the channel before beamforming.

Regarding the practicality of the paper, we acknowledge that the demos we show in this paper are not targeted at actual use cases and we acknowledge that there still exists several limitations that we face towards a commercially-available RF actuation system for soft robots. However, we believe that this work is a large step towards a fully and truly wireless soft robot platform that enables selective actuation of different actuators in a robot inside enclosed spaces. Compared to past works in wireless actuation, including laser [18, 19], NIR [20-23], and microwave [24-25], we demonstrate in our paper coordinated and selected motion between actuators within a larger operation area, which brings capabilities and opportunities to build soft robotic systems towards more complex use cases. We add a clarification sentence in the end of the Introduction in Lines 75-78 on Page 2:

Although the focus of RFact is not in producing actual soft robots that accomplish complex tasks, the demonstration of RFact opens opportunities for using wireless power to actuate soft robots in future work.

[3] The two soft robots developed by the authors are designed with cartoonish frog and turtle appearances. However, it must be said that these two robots do not mimic the movements of frogs and turtles. Frogs jump using their four legs, especially their two hind legs, which is not similar to the soft robot demonstrated by the authors. The authors only showcased a bouncing design, which is not capable of repeated bouncing. Turtles crawl using their four soft legs, but the authors used four wooden sticks to support a turtle-shaped design, which is also quite odd.

We thank the reviewer for their insightful comments. We agree with the reviewer that the two robots do not mimic the movements of frogs and turtles, and we are very sorry for causing the misunderstandings. We have revised and removed any text descriptions that may cause the

misunderstanding and added an explanatory sentence in the demonstration part in Lines 267-268 on Page 10:

The frog and turtle imagery are added solely for demonstrating the jumping and crawling motion of a robot, and are not meant to suggest biologically mimicry of their natural locomotion.

[4] The title of this paper is too broad and does not highlight the technical features of the article. Additionally, radio frequency-driven soft robots have already been adopted by many researchers.

We thank the reviewer for their comments on the paper title. We agree that the title of the paper is too broad. Therefore, we have changed the paper title to “*RFact: Frequency-Selective Actuation of Liquid Crystalline Elastomer Actuators with Radio-Frequency*”, to include our contributions to low-temperature LCE, frequency-selective actuation, and radio-frequency beamforming system.

Past works have studied deploying a radio-frequency power harvesting circuit to provide power for small robots, which falls in the category of power harvesting circuit design, which harvests a few mW of power. In this paper, we study radio-frequency as a direct source of actuation. The advantage of such an approach is that we don't need a power harvesting circuit connected to each actuator, which will cause additional load and bulkiness of the robot while being far less efficient in heating shape memory polymers. Meanwhile, the power input is much larger than power harvesting robots, which translates to larger force output and power density. Past works have studied how to use microwave to efficiently heat the robot, while in this paper we study an efficient and selective actuation scheme to address current challenges in wireless soft robots. We have revised Introduction Lines 50-59 on Page 2 to reflect a clarification.

Prior work has investigated the feasibility of using microwave power to directly heat elastomers through dielectric heating [26, 27] or ferromagnetic alloys through inductive heating [28, 29, 30]. These systems require more than 100 watts of power and tens of seconds to heat materials up to 60°C within a 10-centimeter range as they blindly blast power into the space. Our most recent work shows that through wireless beamforming, soft actuators such as liquid crystal elastomer (LCE) with a conductive layer can be heated more efficiently at a higher speed, even in the presence of occlusions [24, 25]. However, due to the large wavelength of radio frequencies, these previous systems could not be designed to support independent control of multiple actuators, making it impossible for structured coordinated motions between actuators.

[5] The third example presented by the authors involves using an LCE actuator to harvest wireless energy to light up an LED. This demo seems unrelated to soft robots. It merely

demonstrates that wireless power can light up an LED, which has no connection to the deformation and movement of robots.

We thank the reviewer for their comment. Our intention with the LED harvesting example is to show that while the actuator is being actuated, it is also able to deliver electric power to a circuit connected to it. Therefore, while the robot is actuated by the system, operating a low-power device onboard the robot is also possible without the need of carrying a battery, making the robot purely wireless. We do acknowledge that the demonstration here is not directly tied to actuation. To avoid any misunderstandings, we have moved the figures and detailed descriptions of the demo to Supplementary Materials Section 8 and Figure S9.

Figure S9: Supplementary demos: (A) Power harvesting enabled by the actuator during actuation; (B) Actuation of a single actuator inside a pipe.

Nonetheless, we still leave one short paragraph in the main text in the demonstration section in Lines 337-343 on Page 12-13 to discuss the potential needs of deploying small electronics on a soft robots:

Power harvesting during actuation: Except for the ability to be heated and drive the actuation, the resonant conductive trace can also behave as antennas during actuation and support electrical power for on-board electronics. To demonstrate this capability, we connect an LED RF power harvester to the conductive trace on the actuator. Supplementary Movie 3 shows that the LED power harvester lights up when we apply more power to the actuator. The property of simultaneous actuation and power harvesting opens opportunities for the actuation system for more complex tasks that require the soft robot to carry on-board electronics.

[Response to R2]

The authors follow up on a couple of conference proceedings to describe a more detailed version of a more complex system, capable of RF power delivery to a soft robot. Overall the work is timely, well supported by relevant references, and a significant advancement on the field. This reviewer also appreciates the transparency on disclosing the past

conference proceedings and clear delineation on why this is a separate and distinct body of work. Some suggestions for improvement are listed, before the article almost ready for publication, will be ready for a broader audience.

[1] The authors claim this work uses a better actuator, but don't provide a head to head comparison of the actuators, nor a clear benchmarking of the current elastomer. For example, what's the work density of this actuator? What about the power density? How do they compare with other LCEs, or SMAs, or other soft actuators. Even if they don't win on all the metrics, it's still important to the field to show the trade-offs, since this is the one system that is both soft and RF powered.

We thank the reviewer for providing more suggestions on benchmarking to make the paper more solid. We made a more careful analysis of the suggested items and put them in the paper.

To provide a more comprehensive comparison, we evaluated the following results and compared them with an LCE with pure RM257 monomers. All of the other fabrication steps are identical to the current elastomer with 5% Bisphenol-512.

[1.1] The Strain-Temperature curve of the elastomers.

Except for the strain-temperature curve of the current elastomer, we added a curve to evaluate the strain-temperature curve of the LCE elastomer with pure RM257 monomers with DMA testing. The strain-temperature curve shows that under zero load, the strain of both elastomers are the same when reaching the full actuation state. However, since the phase transition temperature of our elastomer is lower, the temperature for our elastomer to reach the minimum strain is also lower. The change is reflected in Results in Lines 134-147 on Page 6.

Figure 2D: Dynamic mechanical analysis (DMA) testing of the LCE with 5% of RM257 replaced by Bisphenol-512.

Figure 2E: DMA of LCE without Bisphenol-512.

Next, dynamic mechanical analysis (DMA) is conducted for the composition in which 5 mol% of RM257 is replaced by Bisphenol A ethoxylate diacrylate ($M_n \sim 512$). The DMA results are shown in Figure 2D. A concentration of 5% Bisphenol A ethoxylate diacrylate is selected for its effective stroke length and transition temperature, which allows for operation in near room temperature environments. An LCE formulation with this composition starts exhibiting contraction when heated from room temperature and completes actuation at approximately 50 °C with 29.6% contraction, which is in line with the stroke length of actuators with an unaltered high T_{ni} LCE chemistry. Moreover, it is similar to the stroke length of natural human muscle and approaches the theoretical maximum of 33.3% for a given 50% pre-stretch.

The DMA result of our LCE is compared with a regular LCE without any Bisphenol-512 monomers. Figure 2E shows that to achieve 10% strain, LCE without Bisphenol-512 needs above 65°C, which is 35°C higher than LCE with 5% Bisphenol-512. Furthermore, LCE with 5% Bisphenol-512 reaches full actuation at around 50°C, which is 40°C less than LCE without Bisphenol-512.

[1.2] The Force-Temperature curve of the elastomers.

We evaluate the force-temperature curve with zero strain on an Instron uniaxial mechanical testing machine. The measurement shows that our actuator achieves higher force output at the same temperature. It also shows while the mechanical performance of our actuator decreases after a 500-cycle test, it still surpasses the regular LCE due to its smaller actuation temperature. The change is reflected in Results in Lines 153-157 on Page 6.

Figure 2F: The force-temperature curve of actuators with 5% Bisphenol-512, actuators after 500 cycles, and unmodified LCE actuator with 0% Bisphenol-512.

The force output of three types of actuators (actuator with 5% Bisphenol-512, actuator (5% Bisp) after actuation of 500 cycles, and LCE with 0% Bisphenol-512) is tested, shown in Figure 2F. The three actuators are cut to the same length with the same cross-sectional areas. LCE actuators with 5% Bisphenol-512 exhibit a shape memory response at a lower temperature, creating a higher force output at lower temperatures in the 30-90°C interval.

[1.3] The Stress-Temperature curve of the elastomers.

The stress-temperature curve is obtained from the force-temperature curve by simply dividing force by cross-sectional area. We also include the error regions of the measurement. It can be seen that LCE with 5% Bisphenol-512 provides higher stress at the same temperature across the temperature region 30-100 degrees Celsius. The change is reflected in Results in Lines 157-162 on Page 6.

Figure 2G-I: The stress-temperature curve of actuators with 5% Bisphenol-512, actuators (5% Bisp) after 500 cycles, and an unmodified LCE actuator with 0% Bisphenol-512. The red and blue shaded areas represent the area within one standard deviation.

By dividing the force output by the cross-sectional areas, Figure 2G-I shows the stress-temperature curve, where the shaded area represents error regions determined by one standard deviation. LCE actuator with 5% Bisphenol-512 provides 350 MPa stress from 25°C to 100°C. Compared to LCE actuator fabricated with pure RM257 [24], the LCE actuator provides 2 times higher stress at 60 °C, and remains to provide 1.5 times higher stress after 500 cycles.

[1.4] The power density of the actuators.

We further evaluate the power density and work density of the actuators by applying a constant 0.5 N load to the actuator and gradually increasing the temperature from room temperature to 100 degrees Celsius. The power density curve shows that a regular actuator provides slightly higher power density than our actuator at the same strain. Note that the temperature for our actuator to achieve the same strain is still lower than the temperature required for a regular actuator. The change is reflected in Results in Lines 163-168 on Page 7.

Figure 2J: The measured power density with a 0.5 N constant load, with regard to strain, for the three types of actuators.

The power and work density of the actuators are measured by applying a 0.5 N (equivalent to a 50 g hanging object) constant load to the actuator and heating it using a constant heat source, by a heat gun. Figure 2J-K show the power and work density during the actuation period of the actuators, with regard to the strain of the actuator. It shows that all actuators have a power density peak between -15% and -20% strain, achieving a maximum of 900 kW/m³ power density for our LCE.

[1.5] The work density of the actuators.

The work density curve is acquired by integrating the power density curve over absolute displacement. It can be seen that although power density of our actuator is smaller, there is no significant difference of work density between the three actuators. The change is reflected in Results in Lines 168-173 on Page 7.

Figure 2K: The measured work density with a 0.5 N constant load, with regard to strain, for the three types of actuators.

Although the power density of LCE with 5% Bisphenol-512 is slightly lower than LCE with 0% Bisphenol-512 and the power density after 500 cycles drops slightly, they still provide similar work density, reaching a total of 20000 kJ/m³ at -25% strain. Since the actuators provide similar work density, the advantage of LCE with 5% Bisphenol-512 actuating at a lower temperature is that it provides opportunities for actuation with less heating time and power.

[1.6] Summary.

To summarize, it can be seen that at the same temperature, our actuator provides larger strain / larger force output than the regular LCE. At the same strain, our actuator provides smaller power density, but the total amount of work is not significantly different. Therefore, our actuator

is fit for applications where input heat power is not high enough to quickly heat the actuator to a very high temperature. In such cases, our actuator is able to achieve faster actuation.

[1.7] Methods.

We describe the test methodologies in the Methods section 4.6 and 4.7, Lines 452-459 on Page 16.

4.6 Force-temperature and stress-temperature test

The force-temperature and stress-temperature tests are conducted using a materials testing machine (Model 5969; Instron). The strain is maintained at 0% during the test, while temperature rises linearly in 30 seconds from 0°C to 100°C.

4.7 Power density and work density test

The power density and work density tests are conducted using a materials testing machine (Model 5969; Instron). The load is maintained at 0.5 N during the test, while temperature rises linearly in 30 seconds from 0°C to 100°C.

[1.8] Comparing with other actuators.

We made a table to compare our proposed LCE with other soft actuators, including SMA, dielectric elastomers, pneumatic actuators, and ionic-diffusion actuators. We have a table in Supplementary Material Table S2 to compare their performance, in page 16 of the supplementary material.

Soft Actuator	Method of Actuation	Actuation strain	Actuation stress
LCE, 5% Bisphenol-512	Heat, above 50°C.	30%	350 kPa
SMA, Ni-Ti spring (2)	Heat, above 120°C.	8%	1 MPa
Dielectric elastomer, DE stack (3)	Voltage, above 4 kV.	30%	50 kPa
Pneumatic, McKibben (4)	Air pump, above 10W.	30%	3 MPa
Ionic diffusion, IPCNC (5)	Voltage, above 4V.	8%	8 kPa

Table S2: Comparison of soft actuators

Table S2 shows the trade-off between five different types of soft actuators. LCE and SMA are actuated by heat. Compared to dielectric elastomer, pneumatic muscles, and ionic diffusion actuators, they are easier to be actuated remotely from a wireless source of heat or power transfer. Compared to SMA springs, LCE provides higher strain but lower stress at a much lower actuation temperature. Dielectric elastomers are actuated at a faster rate, but require high voltage and provide small stress. Pneumatic actuators provide high stress and strain, but require large power input and a tethered air tube. Ionic diffusion actuators can be actuated with small voltage and power, but produces smaller strain and stress.

[2] The system required to drive this seems somewhat complex. A quick search of the power unit showed it's roughly \$10k which would limit adoption. Still, what would be most interesting to this reviewer is a photo at scale showing how much hardware is needed to drive how much actuator. Often in soft robotics some of these details are hidden from the reader, which obscures the message. In the interest of transparency, please add a photo of the entire system required to deliver RF power as described in section 4.8.

We thank the reviewer for the insightful comment. We agree that a photo of the entire system is very helpful for the reader to understand the structure of our whole system. First, we have added images in Figure 4 on Page 11 showing the actuation system and the area of operation.

Figure 4A: The setup of RFact's frequency-selective beamforming system.

We also add descriptions in Result Lines 281-289 on Page 10-12 to describe the setup.

Range of operation: Figure 4A shows our system setup which shows the range of operation of our current implementation. Our current implementation allows an operating range within a $L \times W \times H = 30 \text{ cm} \times 20 \text{ cm} \times 20 \text{ cm}$ cubic area, which is comparable to the operating areas of lasers [18,19] and NIR [20, 21, 22, 23], but allows occlusions and accurate selective actuation of different actuators. The operating area is surrounded by six antennas, each with 10 W maximum transmitting power. The power of each antenna comes from a USRP software-defined radio (SDR) as the signal generator, further passing through an NXP radio-frequency amplifier. We control the frequency, amplitude, and phase of the signals to achieve coherent beamforming by a laptop that is connected to the SDRs.

Moreover, we have also added photos in Supplementary Materials Section 6 and Figure S8.

Figure S8: Setup of our wireless actuation system.

Figure S8 shows a photo of our setup for the actuation platform. The antennas and the robot are inside an enclosure to prevent power leakage in order to comply with FCC regulations. The signal generator is composed of low-noise amplifiers (one for each antenna), a software-defined radio (one RF chain for each antenna), and DC power supplies for the amplifiers.

[3] Do the authors consider the test in Figure 3G to be sufficient to show the actuators are "reliable"? Thinking about possible applications, like the autonomous robot described, that would correspond to about 100 steps. While on this topic, can Figure 4 and the text around it give a clear indication of the speed of the robot in body lengths per second, as is the standard in robotic locomotion demonstrations? Even if slow, it needs to be disclosed for direct comparisons. Coming back to the 100 actuation cycles, this seems insufficient for a robotic application. Can the authors propose an application where 100 cycles would be sufficient? Additionally, the way data is shown in Figure 3G is hard to read, because it's compressed on the vertical axis. What about also showing in the supplemental information a figure where the total contraction is plotted as a fraction of the initial contraction, and have the scale be say 95% - 105%, that would show the reader the variability in actuation much more closely.

We thank the reviewer for their insightful comments on the cycle test and the reliability of the actuator. The 100 cycle test is much less the breaking point of our actuator. In the revised version, we add more results on the cycle test as follows.

We perform cycle test on three independent actuator samples with 20 grams load. All three actuators survive 500 cycles. The time-position curve shows that in the beginning and ending of the 500 cycles, the actuator contraction does not degrade. We continuously keep testing the actuator until its breaking point. The reason for the breaking point is that the resonance of the conductive trace is damaged. We take pictures of the conductive traces before, during, and after

the cycle test to validate. The pictures show that the conductive trace has a capability of self-healing on account of its liquid metal composition. However, due to the alternating-temperature working condition for the cycle test, the gallium-indium that is critical for self-healing gets oxidized. After oxidation, it loses fluidity and the conductive traces fails, causing it to stop resonating under RF stimulation.

The changes are reflected in Figure 3GHI on Page 8 and the text in Results Lines 233-252 on Page 9-10.

Figure 2G: Actuation test of more than 500 cycles shows almost no performance degradation in actuation performance.

Figure 2H: The maximum strain achieved is similar in the first and last 10 cycles.

Figure 2I: Conductive traces before, after 100 cycles, and the broken trace after more than 500 cycles.

For these actuators to reliably operate, they must not degrade either mechanically or electrically when receiving energy from the beamforming system. We conducted a cycle test that included over 500 actuation-cooling cycles powered by the beamforming system to validate the durability of the LCE actuator under wireless actuation and to find the breaking point of the actuator. The cycle test is conducted on three different samples, each tested more than 500 times, and Figure 3G shows the result of the cycle test of one actuator. We use a clay load of 20 grams. All 3 actuators are able to survive 500 cycles. Each heating process includes heating for 60 seconds

by the beamforming system to ensure that the actuator arrives and maintains full actuation, and another 60 seconds to ensure that the actuator is cooled and relaxed. Until 548 cycles, the actuation and cooling speed, as well as the length of the actuator, do not change significantly.

The breaking point of the actuator happens on the conductive trace, where it gets disconnected after a number of cycles. Figure 3I shows three different statuses of the conductive traces after cycles. In the first place, when the trace gets disconnected, EGaIn from the trace flows from the trace to fix the gap, as can be seen in the middle figure. The self-healing procedure also causes some instability in the cycle test. However, during the actuation cycle, the EGaIn gets oxidized under heat and loses fluidity. Therefore, the trace finally gets disconnected and can not be further actuated. In such cases, since the mechanical property of the LCE has not degraded significantly, simply wiping off the original traces and 3D-printing a new trace can re-establish its ability to be actuated.

We also evaluate the mechanical properties of the LCE alone after 500 cycles in response [2]. Since the mechanical properties of the LCE does not degrade much, wiping off the original traces and 3D-printing a new trace would be a way to recover the actuator's resonance. We have added a discussion about future directions to mitigate the problem in Discussion Lines 399-404 on Page 14-15.

Finally, the failure of the conductive elastic ink due to oxidation and mechanical stress over multiple cycles can not be avoided. We note that this limitation can be true of other flexible and stretchable electronics [43] and remains a challenge that remains to be fully addressed. Possible solutions include applying a thin protective layer of elastomer that leads to only a small increase in elastic stiffness without significantly affecting actuation.

We also thank the reviewer for their suggestions of adding speed indicators in the Figures and text. To mitigate the change, we add scale bars, step length for each movement, and time for each step to Figure 4 and the corresponding texts.

Figure 4C: Walking robot with selective actuation. The leg is composed of two actuators, a plastic support, and a foot to create friction. By solely actuating one actuator, the robot moves in the opposite direction. By simultaneously actuating both actuators, the leg returns to the center position.

Figure 4D: Wireless actuation enables a robot to crawl inside a pipe and pick up an object. Selectively actuating the middle actuator can enable crawling movement through the direct friction from the copper feet. Actuating the curling actuator enables lifting of the object.

Jumping demo (Lines 296-297 on Page 12):

The actuator is heated for 90 seconds towards 120 °C until the force it produces balances the magnetic force.

Walking demo (Lines 309-318 on Page 12):

More concisely, we assume that the leg is in the middle position at the beginning of the cycle. To move the robot right, we first actuate the left actuator, which will move the leg left by 1.5 cm for 30 seconds. The bottom of the leg is a foot made of sandpaper that creates friction with the ground. During the leg movement, since the foot does not move relative to the ground, the robot moves right. In the second step, we move the leg by increasing the actuation power of the right actuator and decreasing the actuation power of the left actuator by 30 seconds, which will make the leg move up and back to the initial position. We then wait for the leg to cool down for another round of actuation for 30 seconds. Therefore, the actuation cycle is 90 seconds and step movement is 1.5~cm, which is 1/7 of the robot size.

New crawling demo (Lines 319-336 on Page 12-13):

Crawling inside pipe: *A crawling robot demonstrates how RFact enables selective actuation under the plastic obstruction. The robot is operated in a fully-sealed plastic pipe with 5 mm thickness. Supplementary Movie 6 shows a crawling soft robot driven by the middle actuator. A plastic backbone provides structural stability for the robot. Two copper feet are connected to the plastic backbone. They are cut into jigsaw patterns and bent towards one direction to provide directed friction. Once the middle actuator actuates (30 seconds), it pulls the left foot towards the right by 1.7 cm. When it cools down and relaxes (30 seconds), it pushes the right foot towards outwards by another 1.7 cm. Through such cycles, the robot is able to crawl towards the right. Therefore, the cycle duration is 60 seconds for 3.4 cm of motion, which is 1/3 of the robot size. Another curling actuator is attached to the front of the soft robot. Curling is achieved by applying a layer of Sil-poxy on one side of the actuator, and when heated, the actuator will curl towards the other side. The tip of the curling actuator is covered with adhesive (Sil-poxy in our case). In the first step, we solely actuate the middle actuator, which controls crawling, to make the robot contact the object to be picked. The curling actuator then adheres to the object after the adhesive cures. The motion of lifting the object can be achieved by actuating only the curling actuator. To move the robot after picking the object, we simultaneously actuate the middle and curling actuator, so that the object does not touch the ground while the robot moves.*

[4] The concept is intriguing and likely to make a significant impact on soft robotics. Can the authors comment on how translatable would this RF power delivery be for other soft robotics systems, beyond LCEs?

We thank the reviewer for their valuable comments. The translatability of our RF power delivery method lies in several aspects. We added a paragraph in the discussion section to better describe the details in Lines 363-385 on Page 14:

The demonstration presented here suggests that RFact could potentially be applicable in more practical and complex soft robotic use cases. As an example, RFact allows for power harvesting onboard the soft robot, which enables potential use cases where the soft robot is embedded with sensors and control circuits and is capable of accomplishing more complex tasks.

***Translatability:** The concept and technologies of radio-frequency actuation proposed in RFact are translatable to other soft robotics systems. First, other thermal-responsive elastomers can be combined with resonant conductive patterns to be actuated by the actuation system proposed in RFact. Moreover, shape-memory alloys, if arranged in specific resonant structure, can also be actuated using this approach. Second, beyond thermal-responsive actuators, the frequency-aware beamforming system can be used with an antenna and energy harvesting board as a charging platform, which can be combined with additional types of electrically-responsive actuators. However, it is notable that our current measurement shows the electric power harvested from the actuator is far less than the thermal power, indicating that using electric power harvesters on our system may not create a comparably strong power output. Our platform can also be combined with battery-free wireless soft robots to achieve on-board charging. Finally, we propose an important concept of frequency-selective actuation. Although previous works on NIR light or laser actuation have demonstrated the frequency/wavelength resonance of certain actuators [40,41], RFact is capable of a more fine-grained (100 MHz separation) selective actuation scheme. The concept of frequency-selective actuation can be migrated to other soft robot platforms with other types of wireless power to enable wireless actuation of actuators that are arranged closely.*

[Response to R3]

In this manuscript, Song and Li et al. developed a type of radio frequency actuated LCE for soft robotics. The material and characterizations of the LCE were introduced and applications of the RFact were demonstrated. The reviewer believes that the significance and contribution of this paper are below the baseline for publication in NC in its current

form. Therefore, at least substantial revisions are required before this paper can be considered further for publication.

[1] One of the major comments from the reviewer is the contradictions between these promising statements of the RFact in the Introduction and the weak performances in the demonstrations in later sections. In the Introduction, the authors summarized all the limitations of other soft actuators in mobile, untethered soft robotics and therefore claimed that their RFact can ‘enable more complex movements of soft robots’, ‘enable a series of lightweight soft robotics that can navigate through their environment’. If these can be properly achieved in this paper, the reviewer believes that the significance and contribution of this paper will be sufficient for publication. However, the current demonstration applications in this paper, such as the weak jumping of the frog or the leg motion of the turtle, are far from ‘complex movements’ and ‘navigate through environment’ that the reviewer expects. The reviewer suggests the authors to develop proper soft robotic applications based on their RFact that demonstrate substantial scientific and technological advancements over the state-of-the-art.

We thank the reviewer for pointing out the shortcomings of the demonstrations. The focus of our paper is to introduce a new wireless actuation system (RFact) by microwave actuation, which accomplishes frequency-selective actuation of independent actuators. The demonstrations are used to highlight the different capabilities of RFact. Therefore, we revised the introduction to avoid causing any misunderstanding to the deliverables in the paper. First, we add the following sentence in the introduction to state that the main objective of this paper is to propose a new actuation technology and its corresponding actuator and system design, rather to showing any complex robot movements. The changes are reflected in Lines 73-78 on Page 2.

RFact serves as the first proof-of-concept system towards a wireless actuation platform that achieves a comparable range of mobility as soft robots with on-board batteries [31, 32] – e.g. crawling, hopping, picking up objects, and power harvesting to power electronics. Although the focus of RFact is not in producing actual soft robots that accomplish complex tasks, the demonstration of RFact opens opportunities for using wireless power to actuate soft robots in future work.

Moreover, we also added one more demonstration showing combined selective actuation under occlusions. Supplementary Video S6 shows a crawling robot through a pipe and collecting an object using a curling actuator, also actuated by RFact. It also shows a simple case where the RF can actuate the robot through a narrow enclosure and complete some example tasks. The changes are reflected in Figure 4D and Lines 319-336 on Page 12-13.

***Crawling inside pipe:** A crawling robot demonstrates how RFact enables selective actuation under the plastic obstruction. The robot is operated in a fully-sealed plastic pipe with 5 mm thickness. Supplementary Movie 6 shows a crawling soft robot driven by the middle actuator. A plastic backbone provides structural stability for the robot. Two copper feet are connected to the plastic backbone. They are cut into jigsaw patterns and bent towards one direction to provide*

directed friction. Once the middle actuator actuates (30 seconds), it pulls the left foot towards the right by 1.7 cm. When it cools down and relaxes (30 seconds), it pushes the right foot towards outwards by another 1.7 cm. Through such cycles, the robot is able to crawl towards the right. Therefore, the cycle duration is 60 seconds for 3.4 cm of motion, which is 1/3 of the robot size. Another curling actuator is attached to the front of the soft robot. Curling is achieved by applying a layer of Sil-poxy on one side of the actuator, and when heated, the actuator will curl towards the other side. The tip of the curling actuator is covered with adhesive (Sil-poxy in our case). In the first step, we solely actuate the middle actuator, which controls crawling, to make the robot contact the object to be picked. The curling actuator then adheres to the object after the adhesive cures. The motion of lifting the object can be achieved by actuating only the curling actuator. To move the robot after picking the object, we simultaneously actuate the middle and curling actuator, so that the object does not touch the ground while the robot moves.

Figure 4D: Wireless actuation enables a robot to crawl inside a pipe and pick up an object. Selectively actuating the middle actuator can enable crawling movement through the direct friction from the copper feet. Actuating the curling actuator enables lifting of the object.

We acknowledge that the current demonstrations can not be counted as complex movements or practical use cases. However, it does not mean that the demonstrations show the maximum capability of our system. Rather, by designing more complex robot structures with a complete mechanical design, future works can leverage RFact to produce more complex use cases from a combination of multiple frequency-dependent actuators.

[2] The authors are suggested to summarize the major contributions more clearly in the Introduction. The reviewer noticed that the RFact developed in this paper was extended from many previous works by the authors, as the authors accurately stated in the Introduction and in Results page 4. However, this makes the contributions of this paper unclear, i.e. it reads like this is a paper that simply mixes some ingredients from the authors' different previous works. The authors are suggested to reorganize the Introduction which clearly states the differences between this paper and all related previous works, the challenges solved when adopting these previous works for this paper, and the major contributions of this paper.

We thank the reviewer for their insightful suggestions. We revised the introduction to include a more thorough analysis of the previous works and the difference between our paper with other state-of-the-art solutions. The changes are reflected in Lines 36-59 on Page 1-2.

To combat the limitations caused by tethered robots, efforts have focused on untethered soft robots. Soft robots that carry on on-board batteries [10, 11, 12, 13] are restrained by the size and weight of batteries. To achieve battery-free actuation, researchers have designed actuators that can be actuated under magnetic fields [14, 15, 16] and through inductive heating with coils [17]. The near-field actuation methods are still constrained by the short operating distance, where the driven magnet and coils need to be placed a few millimeters to centimeters away from the actuator. Light sources such as laser [18, 19] and near-infrared [20, 21, 22, 23] actuation can achieve actuation in distance, but they can only operate in line-of-sight. This property not only restrains the operating environment of the robot, but also restricts designs of the robot structures as the actuators must be externally exposed.

This paper builds on our own preliminary work exploring microwave heating as a driving mechanism for soft robots [24, 25]. Radio-frequency is an ideal source for remotely stimulating actuation since it can penetrate through occlusions. Prior work has investigated the feasibility of using microwave power to directly heat elastomers through dielectric heating [26, 27] or ferromagnetic alloys through inductive heating [28, 29, 30]. These systems require more than 100 watts of power and tens of seconds to heat materials up to 60°C within a 10-centimeter range as they blindly blast power into the space. Our most recent work shows that through wireless beamforming, soft actuators such as liquid crystal elastomer (LCE) with a conductive layer can be heated more efficiently at a higher speed, even in the presence of occlusions [24, 25]. However, due to the large wavelength of radio frequencies, these previous systems could not be designed to support independent control of multiple actuators, making it impossible for structured coordinated motions between actuators.

Our contributions include three main parts: a design of a new LCE elastomer with lower phase transition temperature, a study of frequency-selective conductive pattern design and packaging a new actuator, and a frequency-aware beamforming system. The changes are reflected in Lines 60-78 on Page 2.

In this paper, we introduce RFact, a soft robotics platform to achieve accurate, electronics-free wireless actuation that is battery-free, achieving long-range operation to 30 centimeters, the capability to operate through obstructions, and the ability to selectively actuate different actuators. RFact achieves this through a combined design of LCE chemistry, soft actuator integration design, and a radio-frequency beamforming system. First, RFact introduces an LCE actuator that resonates to relatively low-temperature stimulation to fit the context of wireless actuation with less power. Second, RFact exploits the frequency-selective actuation capability of the LCE-based actuators. This is achieved by integrating the actuators with 3D-printed conductive traces that are tuned to resonate at different frequencies. The soft actuator design allows RF heating as well as harvesting power for low-power electronics. Third, RFact proposes a frequency-aware channel interpolation algorithm for beamforming towards a desired actuator location. The frequency-aware beamforming allows actuation at different places and different frequencies through measuring the channel at a few locations in advance.

RFact serves as the first proof-of-concept system towards a wireless actuation platform that achieves a comparable range of mobility as soft robots with on-board batteries – e.g. crawling, hopping, picking up objects, and power harvesting to power electronics. Although the focus of RFact is not in producing actual soft robots that accomplish complex tasks, the demonstration of RFact opens opportunities for using wireless power to actuate soft robots in future work.

[3] The demonstrations in the current version of this paper could benefit from clearer explanations and illustrations of their working principles in Figure 4. For instance, the mechanical design of the frog can be plotted. How is the energy stored, how is this energy released can be drawn more clearly. In the current form, these three demonstrations are restricted in such a small figure, which is completely unnecessary.

We thank the reviewer for their suggestions on making better figures. We agree with the reviewer that Figure 4 lacks sufficient information. Therefore, we have added information in Figure 4 to better describe the working principle of all three demonstrations.

Figure 4B: Jumping robot with wireless actuation. The jumping robot is composed of an actuator and energy storage equipment made of two magnets. In the first step, the actuator is heated with wireless power and produces stress that is applied to the magnet. In the second step, the force is large enough to detach the magnet. When the magnet is detached, the stored energy pushes the jumping robot up.

Figure 4C: Walking robot with selective actuation. The leg is composed of two actuators, a plastic support, and a foot to create friction. By solely actuating one actuator, the robot moves in the opposite direction. By simultaneously actuating both actuators, the leg returns to the center position.

Figure 4D: Wireless actuation enables a robot to crawl inside a pipe and pick up an object. Selectively actuating the middle actuator can enable crawling movement through the direct friction from the copper feet. Actuating the curling actuator enables lifting of the object.

[5] Minor Comments:

[5.1] The last paragraph in Introduction, why this paragraph should exist? Is it for introducing the structure of this paper?

We thank the reviewer for the question. We have removed the paragraph, and revised the first paragraph of the Results section to be a preliminary for all our contributions and results. The changes are reflected in Lines 80-103 on Page 4:

Thermally responsive LCEs typically have phase transition temperatures above 70 °C [32], which requires a large amount of power to achieve full actuation. To actuate LCE actuators towards such a high temperature, high RF input power is used to heat a small region with a small actuator. We aim to activate the LCE actuators with a similar amount of power used in our previous work but with LCE actuators that are of larger size over a larger area. To achieve this, RFact makes use of LCEs that are tuned to have a lower phase transition temperature. The tunable phase transition temperature allows users to choose between high actuation selectivity (co-operate with frequency-selective actuation) and low-power actuation. In particular, we present a method to selectively mix monomers to weaken the interaction between chain bonds by looking into the chemical mechanism of LCE actuation. Specifically we use mixed monomers to weaken the $\pi - \pi$ interaction in the LCE at the nematic phase. Second, we study the resonance of 3D-printed conductive traces. Our past works have shown that using metal coating or conductive-trace coating on LCE can significantly improve actuation speed due to induction heating. This work conducts an extensive study on how the design and fabrication of conductive traces would affect the resonant frequency and heating speed. We then package the LCE with the 3D-printed conductive traces to fabricate actuators that are frequency-selective. Frequency-selective actuators in close spatial regions can be selectively actuated.

Finally, we study the frequency-aware beamforming. Beamforming is achieved through a network of multiple antennas that coherently combines power towards desired locations. The frequency-selective actuation creates new challenges for beamforming since the wireless channel changes with frequencies. We study the standing waves inside the environment and use frequency-aware interpolation to create beamforming weights corresponding to the current operating frequency and the location to be beamformed. The method allows few-shot probes of the channel before beamforming.

[5.2] Figure 1, (A) to (I) captions are wrongly located.

We thank the reviewer for pointing out the mistake. We have corrected the captions in Figure 1 on Page 3.

Figure 1: Overview: (A) LCE-based actuator coated with soft conductive ink for absorbing electromagnetic power and enabling frequency selective heating and actuation. (B) EM waves interact with the conductive elastic patterns, causing induced current to form on the surface and enable inductive heating. (C) On the micro levels, the mesogens which form the LCE structures experience phase transition and become disoriented while being heated, causing the LCE actuator to contract. (D) The wireless actuation platform consists of six antennas that perform multi-antenna distributed beamforming toward target locations. (E) Parallel resonating conductive patterns that is 3D-printed on the actuator can achieve efficient frequency-selective heating. (F) Beamforming achieved by the actuation system can focus wireless power at the desired regions and null the wireless power at undesired regions. (G) A jumper robot is actuated wirelessly by the actuator. We beamform wireless power to the actuator and heat the actuator to over 120°C before the stored mechanic energy is released and the robot jumps. (H) A crawling robot enabled by two separate actuators. The wireless power is beamformed and frequency controlled to actuate each actuator to enable forward and backward locomotion of the soft robot. (I) Passive energy harvesting can be achieved on-robot.

[5.3] The reviewer suggests the authors to enrich the information in all their figures. Please seek suggestions from the senior authors of this paper.

We thank the reviewer for their valuable feedback. We have enriched the information in the Figures. Here is a summary of all the related changes:

1. In Figure 2, we add dashed lines to compare values at the same strain/stress/force.
2. In Figure 3A, we add indicators to highlight important areas.
3. In Figure 3B, we add arrows pointing at LCE and conductive traces.
4. In Figure 3D-F, we marked the resonance frequency of each curve.
5. In Figure 4, we added more descriptive texts on the figures showing the working mechanism and process of the robots.

[5.4] In page 4, the authors stated that the large-scale shape change makes LCE an excellent candidate material for actuators in soft robotics. Please comment on the response rate and actuation period of the LCEs as well.

We thank the reviewer for their comment on adding analysis of the response rate. The time required for actuating our current actuator is 60 second and fully deactuating takes another 60 seconds, in cases with a 20 gram load. With a larger load, the time for actuation can take longer, since there is evidence that with a larger load, the LCE requires a higher temperature to be fully actuated. The result can be read from the cycle test figures in Figure 3G, where each cycle takes 120 seconds, in which the actuation takes 30 seconds and de-actuation takes 90 seconds.

[5.5] Figure 2, the quality of C to E should be improved. The font sizes in these four data plots should be consistent. The unit for temperature °C looks different in D. The reviewer also suggests to enlarge these data plots, the small sizes make seeking information from these plots difficult.

We thank the reviewer for their valuable comments on the quality of the Figures. In the revision, we have improved the quality of all Figures by standardizing and enlarging the figures, improving the organization of the figures, and putting more information into the figures.

[5.6] Page 6, the LCE actuator was tested in cycles, please provide information on how many cycles were tested, either in the text or in the figure.

We thank the reviewer for their comment on extra information for cycle tests. We have followed Reviewer 2's suggestion of adding more cycles and adding more comments on the cycle test. In addition, we have also added more information about the cycle test, including the number of cycles being tested, the breaking point, and the possible solutions to improve the endurance of the elastomer. The change is reflected in Figure 2G on Page 5 and the corresponding text in Lines 234-239 on Page 9:

Figure 2G: Actuation test of more than 500 cycles shows almost no performance degradation in actuation performance.

Figure 2H: The maximum strain achieved is similar in the first and last 10 cycles.

Figure 2I: Conductive traces before, after 100 cycles, and the broken trace after more than 500 cycles.

We conducted a cycle test that included over 500 actuation-cooling cycles powered by the beamforming system to validate the durability of the LCE actuator under wireless actuation and to find the breaking point of the actuator. The cycle test is conducted on three different samples, each tested more than 500 times, and Figure 3G shows the result of the cycle test of one actuator. We use a clay load of 20 grams. All 3 actuators are able to survive 500 cycles.

[5.7] Page 6 bottom, the authors directed the images of the completed actuator to Figure 1A. However, there isn't much information regarding the details of this actuator. More information exists in Figure 3A, B instead.

We thank the reviewer for their comments. We have redirected the images to Figure 3A.

[5.8] Figure 3 D to F. It would be beneficial to provide the frequency at which temperature peaks occur. This is related to the reviewer's comment on enriching the information in these figures.

We thank the reviewer for their suggestions. We have added the frequency at which the peak occurs in Figure 3DEF on Page 8.

Figure 3D: The length of the conductive traces affects the resonant frequency and speed of heating. In particular, we observe that longer traces provide better heating performance, while the length also affects the resonant frequency of the actuator where the maximum heating efficiency is located.

Figure 3E: The roughness of the conductive patterns affects the heating efficiency. Two actuators with different printing configurations, where they present different heating efficiency.

Figure 3F: The resonant frequency and heating efficiency of patterns with different inner-strip distances.

The text also indicates these frequency points in Lines 211-215, 220-222, 228-231 on Page 9.

Results indicate that the maximum temperature is achieved at 65 mm for 2.37 GHz RF stimulation, where the actuator achieves an average of 45 °C. Results also show deviation in the resonant frequency of different actuators, as the optimal resonant frequency is 2.40 GHz for 55 mm-long actuators, 2.47 GHz for 45 mm-long actuators, and 2.43 GHz for 35 mm-long actuators.

The resonant pattern reaches a maximum resonance at 2.39 GHz, 2.40 GHz, and 2.41 GHz for inter-pattern spacing of 0.44 mm, 0.47 mm, and 0.5 mm, respectively.

With the printer nozzle set to 500 μ m above the surface (compared to 200 μ m for regular printing), the temperature increases to an average of 80 °C at 2.39 GHz, shown in Figure 3F. For the pattern with a lower nozzle height, the resonant frequency appears at 2.40 GHz.

RESPONSE TO REVIEWERS

We gratefully thank the reviewers for their efforts in reviewing the paper and their insightful comments, and we appreciate to see the reviewers' recognition of an improvement in the revised manuscript. In the further revision of the manuscript, we have made changes to the introduction and result section to state more clearly our contributions to the field, as well as making the figure presentations more clear to the reader. Our detailed changes are shown as follows.

[1] The author's revisions did not fundamentally enhance the novelty of the article. From a technical perspective, I find the paper feasible and have no objections in that regard. My concerns about the article's innovation primarily stem from the following two aspects:

First, in the field of soft robotics, published works on soft robots consistently demonstrate rich and robust experimental demonstrations. In this regard, the gap between the authors' work and existing literature remains significant. Furthermore, I believe the cartoon depictions of the frog and turtle undermine the article's persuasiveness and demonstrative impact.

We thank the reviewer for their comments. Our experimental demonstrations can be divided into three parts: First, the material characterization shows that with the low phase transition temperature LCE and the actuator can be operated at a lower temperature and with less energy compared to other LCE-based actuators. Second, the radio frequency characterization shows frequency-selective actuation brought by a 3D-printed conductive pattern that is embedded on the surface of the LCE actuator. Third, demonstrations show how frequency-selective actuation of the actuators can be applied to simple robotic prototypes. These include actuation of a jumping robot, tracked actuation of a crawling robot, and combined actuation in which a crawling robot is augmented with an end effector for collecting items.

To enhance the demonstrations, we have included more analysis for the demonstrations by separating Figure 4 into Figures 4 and 5, each demonstrating the speed of movement and the state of the actuators during the full actuation process. We have revised the figures to focus on the actuator design and structure and avoid distraction from the cartoon depictions.

Figure 4: (A) The setup of RFact's frequency-selective beamforming system. (B) A jumping robot enabled by wireless actuation. The jumping robot is composed of an actuator and energy storage equipment made of two magnets. (C) The length of actuator during the actuation period. (D) Screenshot of the actuation process. In the first step, the actuator is heated with wireless power and produces stress that is applied to the magnet. In the second step, the force is large enough to detach the magnet. When the magnet is detached, the stored energy pushes the jumping robot up.

Figure 5: Robotic demonstrations highlighting wireless selective actuation of soft robots. (A) A walking robot enabled by selective actuation. The leg is composed of two actuators, a plastic

support, and a foot to create friction. By solely actuating one actuator, the robot moves in the opposite direction. By simultaneously actuating both actuators, the leg returns to the center position. (B) The length of the left and right actuator, and the position of the leg in one plot, during one step of rightward motion. First, the left actuator is actuated, pulling the foot towards the left. Then, the right actuator is actuated, pulling the foot towards the left in the air. Finally, two actuators are released, while the whole robot moves right. (C) A screenshot of one gait for the robot to move right. (D) A soft robotic walker through a plastic occlusion. Two actuators are selectively activated with varying frequencies, enabling a robot to crawl inside a pipe and pick up an object. (E) Selectively actuating the central horizontal actuator can enable a crawling movement through anisotropically aligned friction from the copper feet. (F) Actuating the curling actuator on the front demonstrates lifting of the object. (G) Simultaneously actuating both actuators enables robot movement while picking the object.

We also revised the texts for the demonstrations correspondingly, which can be seen in Lines 310-366.

Jumping Robot with Single Actuator: To demonstrate this novel robotic platform, we implemented a completely wireless and battery-free jumping motor-spring-latch robot. Inspired by [39], the jumping robot is composed of an LCE actuator, a supporting structure made of plastic, two elastic bands for potential energy buildup, and a pair of 1/16"-wide 1/8"-long magnets (36 kJ/m³, 8300 G) to store potential energy. The structure of the jumping robot is shown in Figure 4B. During the actuation process, wireless power is continuously beamed on the LCE actuator heating and contracting it. The contraction generates a strain on the elastic bands for 90s reaching upwards of 120 °C until the force generated from the elastic potential balances the magnetic latching force. Figure 4C shows the length of the actuator decreases in time from the initial 55 mm to 46.5 mm after a continuous wireless actuation of 70 seconds. After 70 seconds, the actuator length does not change, but the stress induced in the actuator still increases. After full actuation, the stress exceeds the maximum force produced by the magnets, and releases the two magnets, enabling the jumping behavior. While the LCE actuator only weighs 1.5 grams, the jumping robot weighs 45 grams, with the height of jumping reaching 6 cm. Figure 4D shows the screenshot of three processes of the robot actuation, which are the initial state, during actuation, and magnet release. The demonstration indicates the ability of this system to bypass the lower power output of LCE actuation systems, which often need to generate a force over a long period of time, by instead storing the energy for a fast higher power release lifting an object 30 times the weight of the LCE actuator. A video is provided in Supplementary Movie 1.

Walking Robot with Selective Actuation: We show a demonstration of how wireless selective actuation can enable motion on soft robots with two actuators. Figure 5A-C and Supplementary Movie 2 present a walking robot with actuators inducing an antagonistic muscle alignment for movement. For this walker, two LCE actuators resonate at frequencies of 2.36 GHz and 2.4 GHz. By switching the wireless frequency between these two values, it is possible to move the leg upward when both actuators are heated, move down when both are cooled, and move left and right when only the left or right actuator is heated respectively, which is shown in Figure 5A.

Figure 5B further shows the length of the left and right actuator and the position of the foot during the whole actuation cycle, which corresponds to the screenshots in Figure 5C. Initially, the leg is in the middle position at the beginning of the cycle. To move the robot right, we first actuate the left actuator, which will move the leg left by 1.5 cm over 30 seconds. The bottom of the leg is a foot made of sandpaper that creates friction with the ground. During the leg movement, since the foot does not move relative to the ground, the robot moves right. In the second step, we move the leg by increasing the actuation power of the right actuator and decreasing the actuation power of the left actuator for 30 seconds, which will make the leg move up and back to the initial position. We then wait for the leg to cool down for another round of actuation for 30 seconds. This produces an actuation cycle of 90 seconds for one gait with a step movement of 1.5 cm, which is 1/7 of the robot size.

Crawling inside pipe: A crawling robot demonstrates how RFact enables selective actuation under the plastic obstruction. The robot is operated in a fully-sealed plastic pipe with 5 mm thickness. Figure 5D-G and Supplementary Movie 6 show a crawling soft robot driven by the middle actuator. A plastic backbone provides structural stability for the robot and acts a restoring force after actuator cool down. Two copper feet are connected to the plastic backbone. They are cut into jigsaw patterns and bent towards one direction to provide directed anisotropic friction. Once the middle actuator actuates (30 seconds), it pulls the left foot towards the right by 1.7 cm. When it cools down and de-actuates (30 seconds), it pushes the right foot towards the right by another 1.7 cm. Through such cycles, the robot is able to crawl towards the right. Therefore, the cycle duration is 60 seconds for 3.4 cm of motion, which is 1/3 of the robot size. Another curling actuator is attached to the front of the soft robot. Curling is achieved by applying a layer of Sil-poxy on one side of the actuator, and when heated, the stiffness mismatch of the LCE and elastic sil-poxy induces a curling along one direction. The tip of the curling actuator is covered with adhesive (Sil-poxy in this case). In the first step, we solely actuate the middle actuator, which controls crawling, to make the robot move into the position contacting the object to be picked. The curling actuator then adheres to the object after the adhesive cures. The motion of lifting the object can be achieved by actuating only the curling actuator. To move the robot after picking the object, we simultaneously actuate the middle and curling actuator, so that the object does not touch the ground while the robot moves.

[2] Second, the Figures in the article cannot show clear innovation. Is the novelty centered on the low-temperature phase-change liquid crystal material, the radio-frequency (RF) wireless powering method, or the frequency-selective RF powering? Alternatively, does the system's functionality achieve something that cannot be done by other methods? If the latter is the case, the experimental demonstration presented by the author does not seem to be enough good. The low-temperature phase-change material, in my view, is not novel and is widely utilized in liquid crystal research. Regarding the frequency-selective RF powering and wireless powering methods, while the authors have published related studies, these aspects remain underexplored. However, the authors have failed to provide sufficient new data, demonstrations, or detailed analyses to substantiate deeper application-level advancements.

The focus of our paper is on a frequency-selective, power-efficient actuation system for LCE actuators that are completely wireless and do not require line-of-sight for actuation. To achieve this objective, we propose three contributions: (i) a low phase transition temperature LCE recipe that does not compromise mechanical properties when reducing actuation temperature, (ii) a 3D-printed trace that achieves resonance at selective frequencies, and (iii) a frequency-aware beamforming system that can penetrate through obstruction such as a pvc pipe. To better clarify the point, we have revised Lines XXX-XXX in the Introduction section:

RFact is a soft robotics platform that achieves efficient and selective actuation for battery-free thermal actuators. It can operate in a range of 30 centimeters, even with the presence of non-metallic obstructions.

And Lines XXX-XXX in the Results section:

RFact proposes a wireless actuation system that achieves higher efficiency and high selectivity. The efficiency is achieved by designing thermal actuators that actuate at lower temperatures and a wireless multi-antenna beamforming system. The selectivity is achieved by designing frequency-selective actuators and spatially frequency-aware beamforming.

And Lines XXX-XXX in the Results section:

RFact aims at designing an LCE actuator that can produce a larger amount of force output and shape change at a lower and tunable temperature

Additionally, to more clearly explain the advantage and disadvantage of our wireless selective actuation system, we add a table in the supplementary material that compares different actuation methods for soft actuators.

Actuation Method	Power Efficiency (%) From electric to heat	Spatial Selectivity	Frequency Selectivity	Range of Operation	Bulkiness
Wired (6)	~20	Very good	No	Wire length (NLoS)	High (wire tethering)
Batteries (6)	~20	Very good	No	N/A (NLoS)	High (battery weights)
Laser (7)	~30*	Good	Yes	~10 cm (LoS)	Low
Near-field coupling (8)	~30	Bad	No	<10 cm (NLoS)	Low
RF (ours)	~20	Normal	Yes	~30 cm (NLoS)	Low

Table S3: A comparison between different actuation methods for soft thermally-driven actuators. Efficiency of laser actuation is calculated from the multiplying the wall-plug efficiency of laser transmitters and the light-to-heat efficiency of the absorbing materials. LoS=Line-of-Sight. NLoS=None-Line-of-Sight. For wireless actuation methods, NLoS only refers to non-metallic blockages, as electromagnetic fields can not penetrate through metallic blockages.

We compare RFact with existing actuation methods for soft thermally-driven actuators.

Currently, the maximum power efficiency that can be achieved by different actuation systems remains similar. The main challenge in improving the power efficiency for such system is designing more efficient actuators to convert thermal energy into mechanical energy. The spatial selectivity of our system is worse than wired/battery/laser actuation, but when combining spatial selectivity with frequency selectivity, RFact can achieve selective actuation of different actuators that are very nearby. Compared to other wireless actuation methods, RFact achieves non-line-of-sight actuation at a larger range. Compared to wired and battery actuation, wireless actuation has the advantage in low bulkiness and mobility.

We acknowledge that the demonstrations of the paper are not directly targeted at application-level advancements. The key impact of this paper is targeted at bringing the concept of remotely wireless selective actuation of actuators to the field, which can potentially motivate future efforts in developing applications based on our actuation system.